# Comp-Attn: Present-and-Align Attention for Compositional Video Generation

**Hongyu Zhang** [* 1 2]  **Yufan Deng** [* 1 2]  **Shenghai Yuan** [1]  **Xuehan Hou** [1]  **Yian Zhao** [1]
**Peng Jin** [1]  **Chang Liu** [3]  **Jie Chen** [✉ 4 5 1 2]

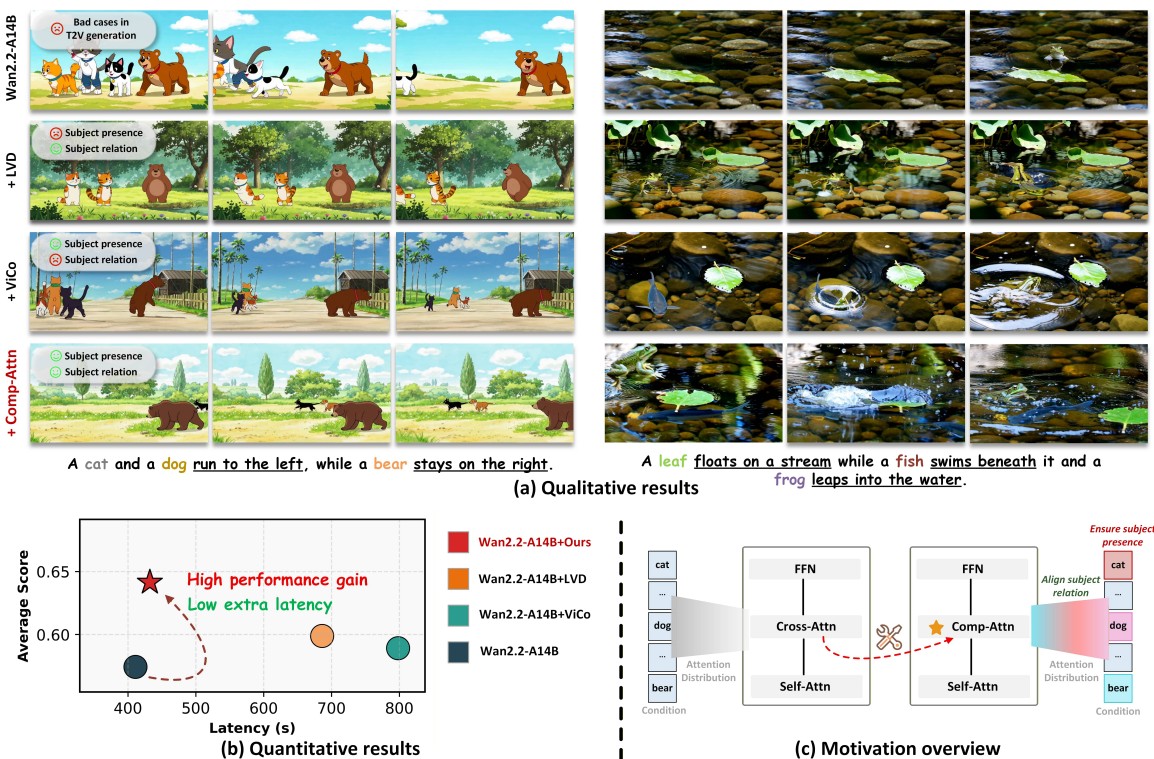

*Figure 1.* We propose **Comp-Attn**, a novel **"Present-and-Align"** paradigm for compositional T2V generation. **(a) Qualitative comparison**, where Comp-Attn effectively addresses both subject presence and inter-subject relation challenges. **(b) Quantitative comparison**, where Comp-Attn achieves significant performance improvement on T2V-CompBench with good efficiency. **(c) Motivation overview**, Comp-Attn injects composition awareness into the condition and attention distribution of the cross-attention layer.

*Equal contribution [1]School of Electronic and Computer Engineering, Peking University, Shenzhen, China [2]AI for Science (AI4S)-Preferred Program, Peking University Shenzhen Graduate School, China [3]Department of Automation and BNRist, Tsinghua University, Beijing, China [4]School of Intelligence Science and Engineering, Harbin Institute of Technology, Shenzhen, China [5]Pengcheng Laboratory, Shenzhen, China. Correspondence to: Jie Chen <chenj@pcl.ac.cn>.

*Proceedings of the 43rd International Conference on Machine Learning*, Seoul, South Korea. PMLR 306, 2026. Copyright 2026 by the author(s).

## Abstract

In the domain of text-to-video (T2V) generation, reliably synthesizing compositional content involving multiple subjects with intricate relations is still underexplored. The main challenges are twofold: 1) Subject presence, where not all subjects can be presented in the video; 2) Inter-subject relations, where the interaction and spatial relationship between subjects are misaligned. Existing methods adopt techniques, such as inference-time latent optimization or layout control, which fail to address both issues simultaneously. To tackle these problems, we propose **Comp-Attn**, a composition-aware cross-attention variant that follows a **"Present-and-Align"** paradigm: it decouples the two challenges by enforcing subject presence at the condition level and achieving relational alignment at the attention-distribution level. Specifically, 1) We introduce Subject-aware Condition Interpolation (SCI) to reinforce subject-specific conditions and

ensure each subject's presence; 2) We propose Layout-forcing Attention Modulation (LAM), which dynamically enforces the attention distribution to align with the relational layout of multiple subjects. Comp-Attn can be seamlessly integrated into various T2V baselines in a training-free manner, boosting T2V-CompBench scores by 15.7% and 11.7% on Wan2.1-T2V-14B and Wan2.2-T2V-A14B with only a 5% increase in inference time. Meanwhile, it also achieves strong performance on VBench and T2I-CompBench, demonstrating its scalability in general T2V and compositional text-to-image (T2I) tasks. Code and models are available at: `https://github.com/Hong-yu-Zhang/Comp-attn`.

## 1. Introduction

Recent text-to-video (T2V) generation has achieved significant breakthroughs, driven by the scaling of data and model sizes (Yu et al., 2022; Esser et al., 2024; Yuan et al., 2024; 2025b; Hong et al., 2022; Wang et al., 2024) as well as the development of novel network architectures (gen, 2024; Pik, 2023; dre, 2024; Dre, 2024; Li et al., 2025b). These advances are driving in a new era of high-quality visual synthesis (Lin et al., 2024; HaCohen et al., 2024; Yang et al., 2025b; Kong et al., 2024; Yuan et al., 2025a; Zhang et al., 2026; Deng et al., 2026). However, existing models consistently struggle to process prompts involving compositional semantics, especially those with multiple subjects and intricate inter-subject relations.

The challenges are twofold: **1) Subject presence**, where not all subjects mentioned in the prompt are accurately represented; **2) Inter-subject relations**, where the interactions and spatial relationships between subjects are often misaligned. These issues lead to subject omission, spatial inconsistencies, and semantic leakage, making compositional T2V generation persistently challenging (Sun et al., 2025).

Existing compositional generation approaches can be broadly categorized into two families: 1) Layout control methods, which utilize object layouts to ensure spatial relationship alignment through masked attention mechanisms (Tian et al., 2024; Wang et al., 2026; Feng et al., 2025; Lian et al., 2024); and (2) Inference-time optimization methods, which perform gradient optimization on visual contents to improve subject presence and adherence to prompts (Chefer et al., 2023; Yang & Wang, 2024). However, the design principles of both methods fall short in simultaneously addressing subject presence and inter-subject relations as illustrated in Figure (a). First, prompts describing multi-entity scenes might introduce semantic interference during conditioning, attenuating the embeddings of

individual subjects (Rassin et al., 2023; Hu et al., 2025). Consequently, layout-control methods that rely solely on attention control can not reliably guarantee subject presence. Second, while inference-time optimization improves equal presence of subjects, the lack of explicit spatial guidance still leads to misaligned inter-subject relations.

To address this, we need to fulfill both challenges simultaneously. The cross-attention layer in diffusion model serves as the interface for injecting guidance semantics, its two key elements are textual conditioning and video-text attention distribution. Specifically, the textual condition determines which subjects are presented in the video, while the video-text attention distribution directly affects the spatial locations and relations of the subjects. ***Thus, can we reengineer these two elements to achieve subject presence and accurate inter-subject relations in a training-free manner?***

Guided by the insights, we propose **Comp-Attn**, a composition-aware cross-attention variant built on a **"Present-and-Align"** paradigm. It tackles the two challenges in a decoupled manner by enforcing subject presence at the conditioning stage and ensuring relational alignment at the attention distribution level. During the conditioning stage, we propose *Subject-aware Condition Interpolation (SCI)* to amplify the presence of each subject. It first estimates the semantic saliency of each subject token within the prompt context to identify potentially suppressed subjects, and then adaptively interpolates subject-specific original semantics based on the saliency scores. At the attention distribution level, we introduce *Layout-forcing Attention Modulation (LAM)* to achieve accurate inter-subject relations. In contrast to previous approaches (Tian et al., 2024; Wang et al., 2026) that impose rigid attention constraints via predefined bounding box (bbox) masks, LAM leverages dynamic attention modulation with intersection over union (IOU) guidance to spatially align the attention distribution of video and subject tokens with the LLM-planned layout, which ensures both accurate spatial relationships and flexibility for diverse subject shapes. Overall, our contributions are summarized as follows:

- We propose Comp-Attn, a novel "Present-and-Align" paradigm to simultaneously ensure subject presence and inter-subject relations for compositional T2V.

- We instantiate the "Present-and-Align" paradigm with two key modules: (1) Subject-aware Condition Interpolation (SCI), which enhances subject-level semantics during conditioning, thereby ensuring the presence of each subject; (2) Layout-forcing Attention Modulation (LAM), which aligns inter-subject relations by spatially attuning video-subject attention distribution to the planned layout.

- Comp-Attn can be seamlessly integrated into ex-

isting frameworks such as Wan (Wang et al., 2025), CogVideoX (Yang et al., 2025b), and VideoCrafter2 (Chen et al., 2024b), significantly boosting compositional generation capability with minimal additional inference overhead. We also demonstrate the broader applications of Comp-Attn, including compositional text-to-image (T2I).

## 2. Related Work

**Text-to-video generation.** Early methods (Zer, 2023; Blattmann et al., 2023; Wang et al., 2023; Khachatryan et al., 2023; Chen et al., 2024b), such as MagicTime (Yuan et al., 2024; 2025b) and AnimateDiff (Guo et al., 2024), integrating temporal modules into the 2D U-Net architecture (Ronneberger et al., 2015), thereby seamlessly extending image generation models to video generation models. Later works, including OpenSora Plan (Lin et al., 2024), CogVideoX (Yang et al., 2025b), LTX-Video (HaCohen et al., 2024), HunyuanVideo (Kong et al., 2024) and Wan (Wang et al., 2025), incorporate a 3D full-attention mechanism, recognized for its superior scalability. This advantage significantly enhances model's ability to capture dynamics in video data, driving rapid progress in T2V (Yuan et al., 2025a; Li et al., 2025b; Chen et al., 2025b; Cai et al., 2025; Zhang et al., 2025b; Xue et al., 2025; Feng et al., 2025). More recently, several studies have integrated diffusion with autoregressive modeling (Chen et al., 2024a; Huang et al., 2026b), which further improves visual quality and video length (Chen et al., 2025a; Teng et al., 2025). However, these foundation models still struggle with compositional prompts, frequently leading to issues like subject omission and misaligned relations.

**Compositional visual generation.** Recent efforts to improve compositionality in visual generation primarily fall into two categories. (1) One line of work focuses on layout-based methods, which introduce explicit layout priors to rearrange the location of subjects using masked or structured attention mechanisms (Lin et al., 2023; Nie et al., 2024; Huang et al., 2026a). While effective in enforcing inter-subject relation constraints, these methods still struggle to ensure the presence of every subject because the semantics of some subjects might have been suppressed during the conditioning phase (Hu et al., 2025), making them difficult to response through attention constraints (Lian et al., 2024; Feng et al., 2025). Moreover, rigid alignment with the layout may constrain the appearance diversity of individual subjects (Tian et al., 2024; Wang et al., 2026). (2) Another line of work explores inference-time optimization strategies to improve subject presence. These methods typically perform test-time latent optimization (Chefer et al., 2023; Hu et al., 2025; Yang & Wang, 2024) by iteratively updating intermediate features using gradients derived from

cross-attention maps, thereby ensuring equal presence for all subjects. However, the inter-subject relations are still hard to align as spatial guidance is not explicitly injected. Moreover, this paradigm significantly increases inference time, limiting its practical utility.

## 3. Method

### 3.1. Overall Framework of Comp-Attn

As illustrated in Figure 2, Comp-Attn is a plug-and-play, composition-aware alternative to the cross-attention mechanism. It follows the "Present-and-Align" paradigm, where SCI module is introduced to enhance subject-specific conditions and ensure the presence of each subject, while LAM module dynamically adjusts the attention distribution to align with the relational LLM layout of multiple subjects. We use GPT-4o-mini (OpenAI, 2024) for efficient subject parsing and layout planning. For simplicity, we only demonstrate the diffusion transformer (DiT) architecture based on cross-attention (Wang et al., 2025). Applying Comp-Attn to the foundation model of the MM-DiT (Yang et al., 2025b) and U-Net (Chen et al., 2024b) architecture follows a similar process (Appendix. A).

### 3.2. Subject-aware Condition Interpolation

Prior studies (Hu et al., 2025; Rassin et al., 2023; Wei et al., 2026) have demonstrated that when prompts contain multiple concepts, the semantic condition of individual subject might be disturbed by other subjects, which leads to subject omission and information leakage. Meanwhile, if the semantic condition of the specific subject is salient, it will exhibit strong and structured attention responses in the video (Helbling et al., 2025). As shown in the top row of Figure 3, we observe that the attention response for the subject "Elliptical mirror" in the video is weak, while the response for "Square window" is excessively strong. This further indicates that the semantic information of certain subjects is being suppressed, failing to ensure its proper presence.

To address this issue, we propose SCI to reinforce the semantic saliency of each subject, ensuring the faithful presence of every subject in the prompt. The underlying idea is intuitive: we aim to adaptively preserve the original semantics of each subject while minimizing the risk of excessive blending with the semantics of other subjects. Specifically, SCI begins with *Subject Encoding* to extract the original semantic embedding of each subject. Next, it performs *Subject Saliency Estimation* to compute saliency scores, which are then used to dynamically perform *Saliency-weighted Interpolation* for each subject. This design effectively restores the original semantics of each subject while preserving their semantic coherence within the context.

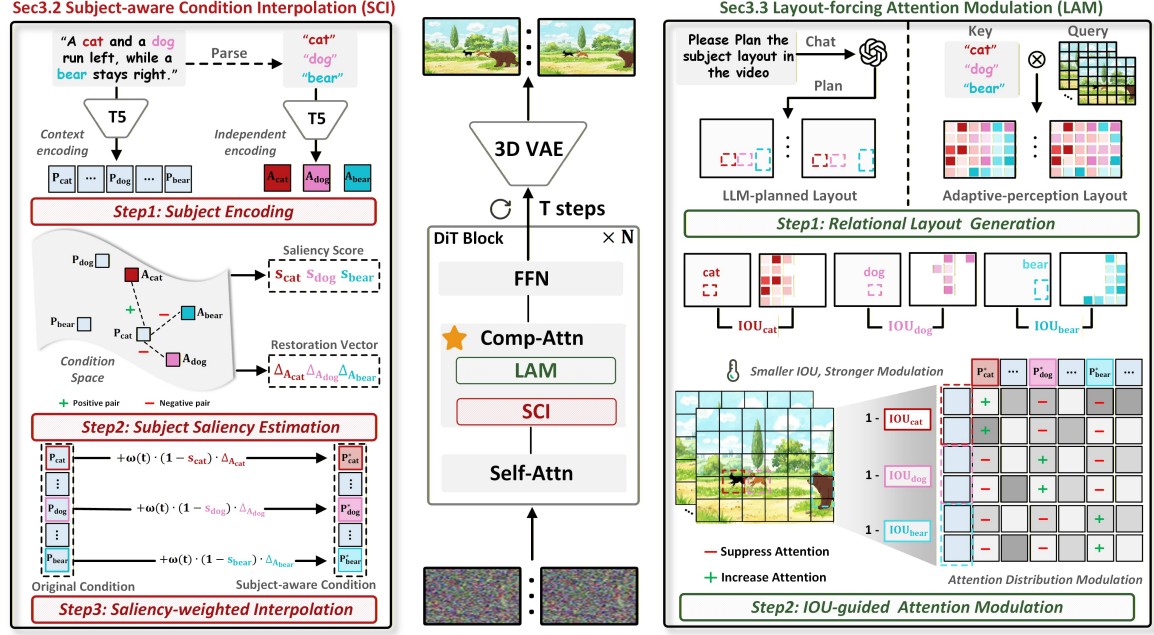

*Figure 2.* **Detailed Architecture of Comp-Attn.** Comp-Attn enhances compositional T2V performance at the cross-attention layer through a "Present-and-Align" paradigm. (1) **Subject-aware Condition Interpolation (SCI)** reinforces subject-specific semantics during conditioning, ensuring the presence of each subject. (2) **Layout-forcing Attention Modulation (LAM)** aligns fine-grained inter-subject relationships by modulating attention distributions.

**Subject Encoding.** For a prompt $S$ containing $M$ subjects $[S_i]_{i=1}^M = [S_1, S_2..., S_M]$, we first obtain the prompt embedding $P = \mathcal{F}(S)$, where $\mathcal{F}(\cdot)$ represents the text encoder. Then the corresponding subject tokens within the context are $[P_i]_{i=1}^M = [P_1, P_2..., P_M]$. Following this, we independently encode each subject without prompt context to obtain the subject anchor embeddings $[A_i]_{i=1}^M = [\mathcal{F}(S_1), \mathcal{F}(S_2)..., \mathcal{F}(S_M)]$, which is considered to contain the pure and original semantics of each subject. As each subject is typically represented by multiple text tokens, we compute the average pooled embeddings $[\overline{P_i}]_{i=1}^M$ and $[\overline{A_i}]_{i=1}^M$ to capture their global semantic representations.

**Subject Saliency Estimation.** For a given subject $k$, we calculate how much of its original semantic information is retained and how much is mixed with the semantics of other subjects. We refer to this as *Subject Saliency*, which are quantified as the degree to which $P_k$ moves away from $[A_i]_{i=1}^M$ (where $i \neq k$) and closer to $A_k$ in the semantic condition space:

$$s_k = \frac{\sum_{i=1}^M 1_{[i=k]} \exp\left(\cos\left(\overline{P_k}, \overline{A_i}\right)/\tau\right)}{\sum_{i=1}^M \exp\left(\cos\left(\overline{P_k}, \overline{A_i}\right)/\tau\right)}. \quad (1)$$

where $s_k$ denotes the subject saliency score for subject $k$ and $\tau$ indicates the temperature factor which is experimentally set to 0.2. A higher saliency $s_k$ indicates that subject $k$ remains closer to its original semantic representation in the feature space while being less affected by interference from other subjects. Similarly, subject saliency scores for all

subjects can be represented as $[s_i]_{i=1}^M$.

**Saliency-weighted Interpolation.** Inspired by the semantic additivity of text embeddings (Hu et al., 2025; Brack et al., 2023), we can manipulate each subject's representation in the feature space to diverge from the semantics of other subjects while converging toward its own semantic essence. As illustrated in Figure 2(a), we can then utilize $[\overline{A_i}]_{i=1}^M$ to define the saliency restoration vector $\Delta_{A_k}$ for subject $k$, which represents the direction for both eliminating semantic confusion from other subjects and reinforcing the subject $k$'s own semantics:

$$\Delta_{A_k} = \sum_{i=1}^M 1_{[i \neq k]}(\overline{A}_k - \overline{A}_i). \quad (2)$$

The saliency restoration vectors for all the subjects $[\Delta_{A_i}]_{i=1}^M$ can be obtained in a similar manner. After that, we can interpolate $[\Delta_{A_i}]_{i=1}^M$ into the original embeddings $[P_i]_{i=1}^M$ according to the subject saliency scores $[s_i]_{i=1}^M$. However, determining the appropriate interpolation strength should also consider the specific denoising time step. Since the early stages of denoising require clearer semantics to capture the basic layout and shape of each subject, while the later stages of denoising focus more on leveraging the context to capture high-level details. To achieve this, we additionally incorporate a time-aware strength attenuated function $\omega(t) = 1 - \frac{t}{T}$ to formulate the final interpolation process:

$$P_i^* = P_i \oplus \left(\omega(t) \cdot (1 - s_i) \cdot \Delta_{A_i}\right). \quad (3)$$

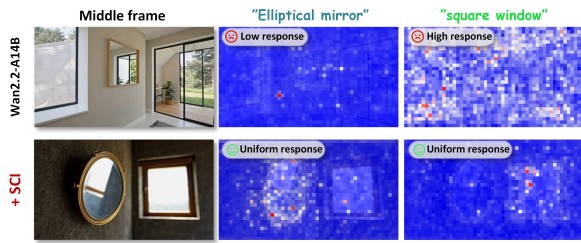

*Figure 3*. **Attention response analysis.** At the 10% timestep, attention scores are averaged across heads. The attention response for "Elliptical mirror" is weak, while that for "Square window" is overly strong, leading to subject absence with Wan2.2-A14B. SCI balances attention responses by restoring the original semantics of subjects, ensuring the presence of each subject.

where $\boldsymbol{P}_i^*$ indicates the interpolated subject embeddings, and $\oplus$ refers to the broadcasting, respectively. Finally, $[\boldsymbol{P}_i]_{i=1}^M$ is replaced by $[\boldsymbol{P}_i^*]_{i=1}^M$ to obtain the subject-aware condition.

After incorporating SCI, the generated content can accurately present all subjects in the prompt, while demonstrating strong and structured correlations in the cross-attention maps as presented in the lower row of Figure 3.

### 3.3. Layout-forcing Attention Modulation

We further propose LAM to align inter-subject relations (e.g., interactions and spatial locations). Unlike previous methods that use hard-masked attention (Tian et al., 2024; **?**) and time-consuming test-time optimization (Lian et al., 2024; Wang et al., 2026), LAM adopts an efficient and soft IOU-guided attention modulation to encourage the attention distribution to align with the LLM-planned layout. Specifically, LAM first conducts *Relational Layout Generation*, which includes the generation of a LLM-planned layout and a adaptive-perception layout. Then, LAM calculates the IOU between the two layouts and uses it as the guiding strength for executing attention modulation.

**LLM-planned Layout Generation.** Given $M$ subjects specified in the prompt, we utilize LLM (OpenAI, 2024) to automatically generate binary layouts for each subject $\boldsymbol{L}^{prior} = \{\boldsymbol{L}_1, \boldsymbol{L}_2, ..., \boldsymbol{L}_M\}$, where $\boldsymbol{L}_i \in \mathbb{R}^{N_{\text{video}} \times 1}$, $N_{\text{video}} = T \times H \times W$ indicates the flattened sequence layout. More details can be found in Appendix D.

**Adaptive-perception Layout Generation.** The adaptive-perception layout is constructed by leveraging the cross-attention map corresponding to each subject. Specifically, given the query matrix of video tokens $\boldsymbol{Q} \in \mathbb{R}^{N_{\text{video}} \times d}$ and the key matrix of text tokens $\boldsymbol{K} \in \mathbb{R}^{N_{\text{text}} \times d}$, the semantic map $\boldsymbol{Attn}_k \in \mathbb{R}^{N_{\text{video}} \times 1}$ associated with the subject $k$ is formulated as:

$$\boldsymbol{Attn}_k = \text{Avg}\left(\mathcal{I}_k\left(\frac{\boldsymbol{Q}\boldsymbol{K}^\top}{\sqrt{d}}\right)\right), \tag{4}$$

where $\mathcal{I}_k(\cdot)$ indexes the attention weights corresponding to the subject $k$, and the operation $\text{Avg}(\cdot)$ computes the mean across the selected attention scores. We further adopt the mean thresholding method in (Helbling et al., 2025) to obtain the adaptive-perception layout $\boldsymbol{L}_k^{adapt} \in \mathbb{R}^{N_{\text{video}} \times 1}$ for subject $k$:

$$\boldsymbol{L}_k^{adapt} = \delta\big(\boldsymbol{Attn}_k \geq \text{Mean}(\boldsymbol{Attn}_k)\big), \tag{5}$$

where $\delta(\cdot)$ represents the threshold function. Similarly, we obtain the layouts for other subjects: $\boldsymbol{L}^{adapt} = \{\boldsymbol{L}_1^{adapt}, \boldsymbol{L}_2^{adapt}, ..., \boldsymbol{L}_M^{adapt}\}$.

**IOU-guided Attention Modulation.** We achieve the dynamic alignment of attention distribution with the LLM-planned layout through attention bias modulation, which can be formulated as:

$$\boldsymbol{Attn}^{\text{mod}} = \text{softmax}\left(\frac{\boldsymbol{Q}\boldsymbol{K}^\top + \sum_{k=1}^M \boldsymbol{L}_k \odot \boldsymbol{Bias}_k}{\sqrt{d}}\right), \tag{6}$$

$$\boldsymbol{Bias}_k = \boldsymbol{G}_k(\boldsymbol{Q}, \boldsymbol{K}) \odot \boldsymbol{E}_k(\boldsymbol{Q}, \boldsymbol{K}), \tag{7}$$

where $\boldsymbol{G}_k(\boldsymbol{Q}, \boldsymbol{K})$ determines whether to increase or decrease attention, and $\boldsymbol{E}_k(\boldsymbol{Q}, \boldsymbol{K})$ controls the strength of the modulation.

The core idea of $\boldsymbol{G}_k(\boldsymbol{Q}, \boldsymbol{K})$ is to enhance the interaction between the layout region of subject $k$ and its corresponding text tokens, while simultaneously weakening the interactions with text tokens of other subjects, which is conducted by:

$$\boldsymbol{g}_i^+ = \max(\boldsymbol{Q}\boldsymbol{K}^\top) - \boldsymbol{Q}\boldsymbol{K}^\top, \tag{8}$$

$$\boldsymbol{g}_i^- = \min(\boldsymbol{Q}\boldsymbol{K}^\top) - \boldsymbol{Q}\boldsymbol{K}^\top, \tag{9}$$

$$\boldsymbol{G}_k(\boldsymbol{Q}, \boldsymbol{K})[x, y] = \begin{cases} \boldsymbol{g}_i^+[x, y], & \text{if } y \in \boldsymbol{S}_k, \\ \boldsymbol{g}_i^-[x, y], & \text{if } y \in \boldsymbol{S}_l, k \neq l \\ 0, & \text{otherwise}, \end{cases} \tag{10}$$

where $\boldsymbol{g}_i^+$ and $\boldsymbol{g}_i^-$ denote the positive term applied to subject $k$ and the negative term applied to other subjects, respectively. $[x, y]$ specifies the indices of the attention matrix.

We then determine the modulation strength through $\boldsymbol{E}_k(\boldsymbol{Q}, \boldsymbol{K})$. Instead of simply relying on timesteps or object sizes (Kim et al., 2023), we adopt a more flexible approach that adjusts the modulation based on the IoU between the current attention distribution mask $\boldsymbol{L}_k^{adapt}$ and the planned layout $\boldsymbol{L}_k$:

$$\boldsymbol{IoU}_k = \frac{\sum_{i=1}^N \min(\boldsymbol{L}_k^{adapt}[i], \boldsymbol{L}_k[i])}{\sum_{i=1}^N \max(\boldsymbol{L}_k^{adapt}[i], \boldsymbol{L}_k[i])}, \tag{11}$$

$$\boldsymbol{E}_k(\boldsymbol{Q}, \boldsymbol{K}) = 1 - \boldsymbol{IoU}_k. \tag{12}$$

This approach establishes a flexible feedback mechanism, allowing the model to balance original attention and layout-region attention, avoiding excessive modulation that may reduce content diversity.

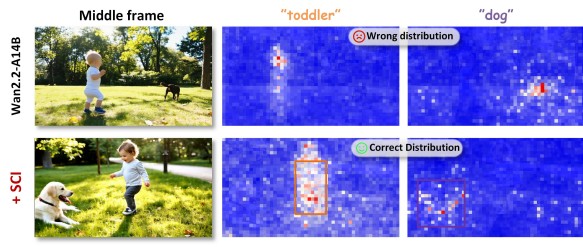

*Figure 4.* **Attention distribution analysis.** The incorrect spatial attention distribution in Wan2.2-A14B causes positional errors, while LAM adjusts it to accurately reflect inter-subject relationships. The colored boxes represent the LLM layout.

As shown in Figure 4, LAM spatially aligns attention distribution with multiple subjects while maintaining visual quality, further demonstrating the effectiveness of our approach in ensuring inter-subject relationships.

# 4. Experiment

## 4.1. Experimental Setups

**Implementation Details.** We seamlessly incorporate Comp-Attn into four representative video generation models: (1) Wan2.1 and Wan2.2 (Wang et al., 2025), which utilize a DiT-based framework with cross-attention for integrating text features; (2) CogVideoX (Yang et al., 2025b), which employs a MM-DiT architecture; (3) VideoCrafter2 (Chen et al., 2024b), built on a UNet-style backbone. We further apply Comp-Attn to FLUX (flu, 2024) to demonstrate its generalizability in T2I tasks. Comp-Attn is applied during the first 20% of the denoising timesteps for all models. We use the GPT-4o-mini (OpenAI, 2024) API to generate planned layouts, each taking around 5 seconds to response (Appendix D). All videos are generated on NVIDIA H100 GPUs. More details can be found in Appendix A.

**Benchmark and Evaluation Metrics.** We employ T2V-CompBench (Sun et al., 2025) and VBench (Huang et al., 2024) for evaluation. T2V-CompBench is a large-scale benchmark designed to evaluate compositional T2V performance across seven dimensions: consistent attribute binding, dynamic attributes, spatial relationships, motion binding, action binding, object interactions, and generative numeracy. Each dimension includes 200 prompts. VBench is used as a supplementary benchmark to verify whether the model can balance generation performance in terms of both general video quality and semantics. It consists of 16 dimensions, each with approximately 100 prompts, which can be grouped into two major categories: quality and semantics. We additionally employ T2I-CompBench (Huang et al., 2025) to evaluate the performance of extending Comp-Attn to compositional T2I tasks.

**Evaluated Models.** We compare our approach against two groups of baseline models: (1) Basic T2V founda-

tion models, which include Gen-3 (gen, 2024), Dreamina1.2 (Dre, 2024), Open-Sora1.2 (Zheng et al., 2024), T2V-Turbo-v2 (Li et al., 2025a), and SkyReels-v2 (Chen et al., 2025a); (2) Compositional T2V models, which include layout control methods such as LVD (Lian et al., 2024), VideoTetris (Tian et al., 2024), and the inference-time optimization technique Vico (Yang & Wang, 2024). We reproduce LVD and Vico, which do not require training data, on each baseline foundation model to ensure a fair comparison. We present the implementation details in Appendix B.

## 4.2. Quantitative Comparisons

**Compositional T2V Evaluation.** As shown in Table 1, Comp-Attn has achieved significant performance improvements of 14.77%, 29.84%, 15.71%, and 11.70% on the baseline models VideoCrafter2, CogVideoX-5B, Wan2.1-14B, and Wan2.2-A14B, respectively. Moreover, it outperforms compositional T2V methods such as Vico and LVD across all baseline models. Notably, CompAttn achieved its most significant gains on metrics demanding subject presence and inter-subject relations (e.g., spatial relationship and interaction), with improvements of 23.01% and 29.59% on CogVideoX, and 18.64% and 19.16% on Wan2.1-14B. This result solidly validates the effectiveness of the "Present-and-Align" paradigm.

**Compositional T2I Evaluation.** Table 3 indicates that applying Comp-Attn to FLUX enhances its compositional image generation capability, achieving a 5.36% average improvement on T2I-CompBench. Notably, it also outperforms state-of-the-art training-based methods, SiamsLayout (Zhang et al., 2025a) and HybridLayout (Wu et al., 2025). This strongly validates Comp-Attn's extensibility to broader visual generation tasks.

**General T2V Evaluation.** We investigate whether Comp-Attn impacts general video generation quality. Table 2 shows that our method significantly improves the VBench semantic score, benefiting conventional video generation by handling complex semantics (e.g., multiple objects and spatial relations). At the same time, Comp-Attn maintains baseline performance on the quality score, demonstrating its ability to inject composition-awareness without compromising the foundation model's core representation. In contrast, models like LVD and Vico show a noticeable drop in quality score, likely due to train-test gaps introduced by latent optimization, which interfere with the baseline models' core capabilities.

**Inference Time Analysis.** To assess inference efficiency, we evaluate Comp-Attn against other models by generating videos with a resolution of 81×480×832. The comparison leverages the average inference latency in T2V-CompBench, which incorporates the overhead of calling the GPT-4o-mini API. Table 4 highlights that Comp-Attn achieves a mere 5.1% increase in inference time compared to the baseline

*Table 1.* Quantitative comparison results. Best/2nd best scores are **bolded**/underlined. † indicates commercial models.

| Model | Con-attr | Dyn-attr | Spatial | Motion | Action | Interact | Num | Avg |
|---|---|---|---|---|---|---|---|---|
| Gen-3 † (gen, 2024) | 0.5980 | 0.0687 | 0.5194 | 0.2754 | 0.5233 | 0.5906 | 0.2306 | 0.4008 |
| Dreamina † 1.2 (Dre, 2024) | 0.6913 | 0.0051 | 0.5773 | 0.2361 | 0.5924 | 0.6824 | 0.4380 | 0.4604 |
| Kling-1.0 † (Kling, 2024) | 0.6931 | 0.0098 | 0.5690 | 0.2562 | 0.5787 | 0.7128 | 0.4413 | 0.4658 |
| T2V-Turbo-V2 (Wang et al., 2023) | 0.6723 | 0.0127 | 0.5025 | 0.2556 | 0.6087 | 0.6439 | 0.3261 | 0.4317 |
| Open-Sora 1.2 (Zheng et al., 2024) | 0.5639 | 0.0189 | 0.5063 | 0.2468 | 0.4833 | 0.5039 | 0.3719 | 0.3850 |
| VideoTetris (Tian et al., 2024) | 0.6211 | 0.0104 | 0.4832 | 0.2249 | 0.4939 | 0.6578 | 0.3467 | 0.4054 |
| MAGI-1-24B | 0.7942 | 0.0419 | 0.5918 | 0.3259 | 0.7061 | 0.7365 | 0.4486 | 0.5207 |
| SkyReels-v2-13B | 0.8339 | 0.0628 | 0.6325 | 0.3467 | 0.7422 | 0.7502 | 0.4218 | 0.5414 |
| VideoCrafter2 (Chen et al., 2024b) | 0.6182 | 0.0103 | 0.4838 | 0.2259 | 0.5030 | 0.6365 | 0.3330 | 0.4015 |
| + LVD (Lian et al., 2024) | 0.6628 | 0.0097 | 0.5487 | 0.2627 | 0.5637 | 0.6713 | 0.3561 | 0.4393 |
| + ViCo (Yang & Wang, 2024) | 0.6539 | 0.0186 | 0.5120 | 0.2154 | 0.5481 | 0.6957 | 0.3618 | 0.4294 |
| + Ours | 0.7036 | 0.0229 | 0.5718 | 0.2519 | 0.5809 | 0.7296 | 0.3652 | 0.4608 |
| △-Baseline | +14.25% | +122.3% | +18.19% | +11.51% | +15.49% | +14.63 | +9.67% | +14.77% |
| CogVideoX-5B (Yang et al., 2025b) | 0.6164 | 0.0219 | 0.5172 | 0.2658 | 0.5333 | 0.6069 | 0.3706 | 0.4189 |
| + LVD (Lian et al., 2024) | 0.6976 | 0.0380 | 0.5906 | 0.3256 | 0.6402 | 0.6546 | 0.3867 | 0.4762 |
| + ViCo (Yang & Wang, 2024) | 0.7048 | 0.0191 | 0.5450 | 0.2834 | 0.6039 | 0.6978 | 0.3952 | 0.4642 |
| + Ours | 0.7839 | 0.0424 | 0.6362 | 0.4157 | 0.6824 | 0.7865 | 0.4606 | 0.5439 |
| △-Baseline | +27.17% | +93.61% | +23.01% | +56.40% | +27.96% | +29.59% | +24.28% | +29.84% |
| Wan2.1-14B (Wang et al., 2025) | 0.8256 | 0.0751 | 0.6073 | 0.3062 | 0.7439 | 0.7144 | 0.4780 | 0.5358 |
| + LVD (Lian et al., 2024) | 0.8432 | 0.0689 | 0.6425 | 0.3819 | 0.7602 | 0.7570 | 0.4867 | 0.5629 |
| + ViCo (Yang & Wang, 2024) | 0.8125 | 0.0586 | 0.6294 | 0.3427 | 0.7930 | 0.7783 | 0.4708 | 0.5551 |
| + Ours | 0.8614 | 0.0980 | 0.7205 | 0.4249 | 0.8344 | 0.8513 | 0.5508 | 0.6200 |
| △-Baseline | +4.34% | +30.49% | +18.64% | +38.77% | +12.17% | +19.16 | +15.23% | +15.71% |
| Wan2.2-A14B (Wang et al., 2025) | 0.8394 | 0.1261 | 0.6618 | 0.3654 | 0.7526 | 0.7902 | 0.4839 | 0.5742 |
| + LVD (Lian et al., 2024) | 0.8431 | **0.1421** | 0.6973 | 0.3958 | 0.8039 | 0.8224 | 0.4870 | 0.5988 |
| + ViCo (Yang & Wang, 2024) | 0.8539 | 0.0927 | 0.6725 | 0.3761 | 0.7950 | 0.8306 | 0.5031 | 0.5891 |
| + Ours | **0.8726** | 0.1315 | **0.7244** | **0.4708** | **0.8539** | **0.8598** | **0.5769** | **0.6414** |
| △-Baseline | +3.96% | +4.28% | +9.46% | +28.84% | +13.46% | +8.81% | +19.22% | +11.70% |

*Table 2.* Quantitative comparison on VBench.

| Method | Wan2.2-A14B | | | CogVideoX-5B | | |
|---|---|---|---|---|---|---|
| | Total | Quality | Semantic | Total | Quality | Semantic |
| Baseline | 0.8518 | 0.8603 | 0.8176 | 0.8201 | 0.8272 | 0.7917 |
| + ViCo (Yang & Wang, 2024) | 0.8297 | 0.8310 | 0.8245 | 0.8113 | 0.8125 | 0.8064 |
| + LVD (Lian et al., 2024) | 0.8360 | 0.8370 | 0.8319 | 0.8055 | 0.8030 | 0.8153 |
| + Ours | **0.8574** | **0.8628** | **0.8357** | **0.8304** | **0.8329** | **0.8205** |
| △-Baseline | +0.66% | +0.29% | +2.21% | +1.26% | +0.69% | +3.64% |

*Table 3.* Quantitative results on T2I-CompBench.

| Method | Spatial | Color | Shape | Texture | Numeracy | Avg |
|---|---|---|---|---|---|---|
| FLUX | 0.2863 | 0.7407 | 0.5718 | 0.6922 | 0.6185 | 0.5819 |
| +SiamLayout (Zhang et al., 2025a) | **0.3835** | 0.7503 | 0.5304 | 0.6035 | 0.5467 | 0.5629 |
| +HybridLayout (Wu et al., 2025) | 0.3208 | **0.7960** | 0.5870 | 0.6882 | **0.6274** | 0.6039 |
| + Ours | 0.3642 | 0.7714 | **0.6021** | **0.7056** | 0.6221 | **0.6131** |
| △-Baseline | +27.21% | +4.14% | +5.30% | +1.94% | +0.58% | +5.36% |

*Table 4.* Inference latency comparison on a single H100 GPU.

| Model | Wan2.2-A14B | + LVD | + ViCo | + Ours |
|---|---|---|---|---|
| Latency (s) | 411 | 686 | 798 | 432 |

motion directions accurately aligned. Similarly, the example on the right of Figure 5 illustrates that our method is the only one capable of accurately depicting three subjects while capturing the fine-grained relationships among them. These results fully demonstrate the effectiveness of the Present-and-Align paradigm of Comp-Attn compared to other compositional paradigms like LVD and ViCo. Additional qualitative results are included in the Appendix E.

*Table 5.* Ablation study of SCI and LAM modules.

| Method | Con-attr | Spatial | Interact | Action | Time |
|---|---|---|---|---|---|
| CogvideoX-5B | 0.6164 | 0.5172 | 0.6069 | 0.5333 | **93s** |
| w/ SCI | 0.7182 | 0.5807 | 0.7124 | 0.6122 | 95s |
| w/ LAM | 0.7346 | 0.6040 | 0.7396 | 0.6361 | 104s |
| **w/ SCI & LAM** | **0.7839** | **0.6362** | **0.7865** | **0.6824** | 106s |
| Wan2.2-A14B | 0.8394 | 0.6618 | 0.7902 | 0.7526 | **411s** |
| w/ SCI | 0.8613 | 0.6840 | 0.8357 | 0.8022 | 413s |
| w/ LAM | 0.8595 | 0.7079 | 0.8296 | 0.8213 | 430s |
| **w/ SCI & LAM** | **0.8726** | **0.7244** | **0.8598** | **0.8539** | 432s |

model Wan2.2-A14B while delivering substantially better performance than methods such as LVD and ViCo.

### 4.3. Qualitative Comparisons

Figure 5 demonstrates the advantages of our method in generating complex compositional content. Comp-Attn faithfully represents each subject in the prompt and correctly reflects the spatial relationships and motion directions between the subjects. For instance, in the example on the left of Figure 5, the red car and the cyclist are consistently present in every frame, with their relative positions and

### 4.4. Ablation Study

**Ablation Results on SCI and LAM.** Table 5 shows that both core components of Comp-Attn contribute positively to the final performance. Their synergy further enhances the

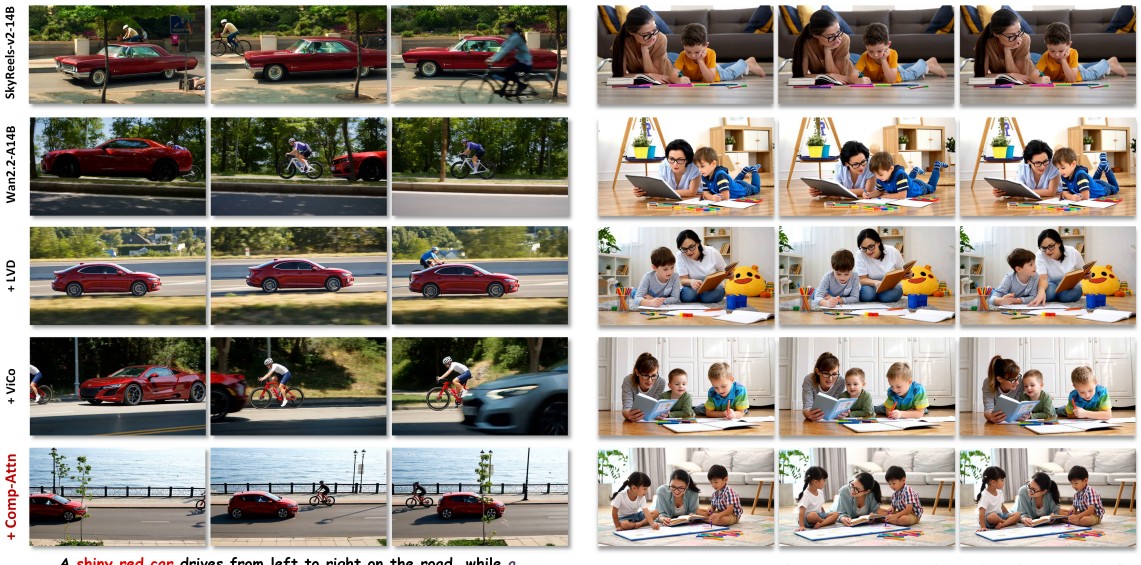

*Figure 5.* **Qualitative comparison on generating compositional contents.** Comp-Attn achieves superior performance in both subject presence and inter-subject relations, surpassing other compositional generation paradigms as well as powerful video foundation models.

*Table 6.* Individual analysis on SCI and LAM modules.

| Method | Con-attr | Spatial | Interact | Action | Time |
|---|---|---|---|---|---|
| Wan2.2-A14B | 0.8394 | 0.6618 | 0.7902 | 0.7526 | 411s |
| *Subject presence* | | | | | |
| + SEGA (Brack et al., 2023) | 0.8539 | 0.6405 | 0.8081 | 0.7724 | 706s |
| + ToMe (Hu et al., 2025) | 0.8446 | 0.6673 | 0.8208 | 0.7652 | 744s |
| **w/ SCI (Ours)** | **0.8613** | **0.6840** | **0.8357** | **0.8022** | **413s** |
| *Inter-subject relation* | | | | | |
| + DreamRunner (Wang et al., 2026) | 0.8178 | 0.6906 | 0.8144 | 0.7930 | 472s |
| + VideoGrain (Yang et al., 2025a) | 0.8261 | 0.6813 | **0.8350** | 0.8028 | 495s |
| **+ LAM (Ours)** | **0.8595** | **0.7079** | 0.8296 | **0.8213** | **430s** |

*Table 7.* Ablation study on the IOU-guidance strategy in LAM.

| Method | Con-attr | Spatial | Interact | Action | Time |
|---|---|---|---|---|---|
| w/o IOU-guidance | 0.8364 | 0.6928 | 0.7981 | 0.7850 | **428s** |
| **LAM (Ours)** | **0.8595** | **0.7079** | **0.8296** | **0.8213** | 430s |

results, which is particularly evident in metrics such as "spatial relationship" and "interaction," where subject presence and inter-subject relations are simultaneously emphasized.

**Individual analysis on LAM and SCI.** Beyond Comp-Attn's overall gains, Table 6 compares SCI and LAM with prior methods. For subject presence, SCI surpasses ToMe (Hu et al., 2025) and SEGA (Brack et al., 2023) with higher efficiency, suggesting that condition-level optimization is gentle yet effective. For inter-subject relations, LAM outperforms VideoGrain (Yang et al., 2025a) and Dream-Runner (Wang et al., 2026). This demonstrates that LAM's soft attention distribution adjustment mechanism is superior to hard masked attention (Wang et al., 2026). Additionally, LAM's IOU-guidance mechanism offers greater flexibility and robustness compared to modulation intensity control based on time steps (Yang et al., 2025a).

**Effect of IOU-guidance in LAM.** Table 7 shows that removing IoU-guidance from LAM significantly degrades

performance, underscoring the proposed IoU-guided attention modulation's adaptability.

**Effect of Subject Saliency Estimation in SCI.** We further discuss the role of subject saliency estimation in SCI. Experiments in Appendix C show that replacing the subject saliency score with constant interpolation coefficients leads to significant performance drops. This further demonstrates that subject saliency estimation adaptively restores original semantics, preventing disruption of contextual semantics.

**Robustness of LLM Layout planning.** In Appendix F, we discuss the error rate of LLM Layout Planning (less than 5%), techniques to reduce it (to below 2%), and how Comp-Attn handles situations when errors occur.

## 5. Conclusion

This paper presents **Comp-Attn**, a training-free and model-agnostic framework for compositional T2V generation. Based on the "Present-and-Align" paradigm, it addresses challenges of "subject presence" and "inter-subject relation" by injecting composition-awareness into the conditioning process and attention distribution of cross-attention. Specifically, **1) Subject-aware Condition Interpolation (SCI)** strengthens subject-specific conditions to ensure subject presence; **2) Layout-forcing Attention Modulation (LAM)** dynamically adjusts attention to align with relational layouts. Seamlessly integrable with frameworks like Wan, CogVideoX, and VideoCrafter2, Comp-Attn boosts performance with minimal inference overhead. It also generalizes to compositional T2I tasks, showcasing broad applicability and adaptability. Comp-Attn provides an efficient and robust solution for compositional visual generation.

## Acknowledgements

This work was supported in part by the New Generation Artificial Intelligence-National Science and Technology Major Project (No. 2025ZD0122702), the Shenzhen Medical Research Funds in China (No. B2302037), the Natural Science Foundation of China (No. 61972217, 32071459, 62176249, 62006133, 62271465), the AI for Science (AI4S)-Preferred Program, Peking University Shenzhen Graduate School, China, and the Shenzhen Loop Area Institute under grant FPF10120250014.

## Impact Statement

This paper presents work aimed at advancing compositional visual generation. There are many potential societal consequences of our work, none of which we feel need to be specifically highlighted here.

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

# Comp-Attn: Present-and-Align Attention for Compositional Video Genneration
## Appendix

## A. Adapt Comp-Attn to Other Models

**Wan2.2 (Wang et al., 2025).** Wan2.2-A14B and Wan2.1-14B are largely consistent in terms of model architecture. The key difference lies in Wan2.2-A14B adopting a mixture-of-experts architecture, where two DiT models with identical parameter sizes and architectures are used to handle high-noise and low-noise latents, respectively. Therefore, we can seamlessly integrate Comp-Attn without modifying any of its components, simply by switching between the two experts according to the original timestep settings of Wan2.2.

**CogVideoX (Yang et al., 2025b).** CogVideoX is built on the MM-DiT architecture, which requires slight modifications to Comp-Attn for proper adaptation. For SCI, we perform subject-aware interpolation on the condition and concatenate it with the video tokens before feeding them into the network. For LAM, CogVideoX employs multimodal self-attention, where the attention distribution is divided into four regions: video-video, video-text, text-video, and text-text (Cai et al., 2025). We modulate the attention probe specifically within the video-text region to avoid affecting the attention interactions in other regions. Specifically, we inject a modulation bias term into the video-text region of the multimodal attention map to achieve a similar effect as cross-attention in the DiT architecture.

**VideoCrafter2 (Chen et al., 2024b).** Comp-Attn can be seamlessly integrated into U-Net-based models, such as VideoCrafter2, with minimal adaptation effort. Specifically, SCI can be directly added to the conditioning stage of VideoCrafter2, enhancing the subject semantics of CLIP text embeddings (Radford et al., 2021). For LAM, we choose to perform attention modulation on the intermediate layers of the encoder and decoder, as these layers are semantically rich in the U-Net architecture.

**FLUX (flu, 2024).** It is also a model based on the MM-DiT architecture, where we apply Comp-Attn following the approach used in CogVideoX. The differences lie in two aspects: (1) the visual input consists of image tokens, and (2) it employs both dual-stream and single-stream blocks to achieve different multimodal feature modulations. However, the specific attention interaction rules remain unchanged.

## B. Baseline Implementation Details

For a fair comparison, the quantitative performance of the following models is directly taken from the T2V-CompBench leaderboard (Sun et al., 2025): Gen-3 (gen, 2024), Dreamina1.2 (Dre, 2024), Kling-1.0 (Kling, 2024), VideoCrafter (Chen et al., 2024b), CogVideoX-5B (Yang et al., 2025b), Open-Sora1.2 (Zheng et al., 2024), T2V-Turbo-v2 (Li et al., 2025a), and VideoTetris (Tian et al., 2024). To expand the scope of validation, we reproduce two representative autoregressive diffusion paradigms, MAGI-1-24B (Teng et al., 2025) and SkyReels-V2-13B (Chen et al., 2025a), on the T2V-CompBench. Meanwhile, we also reproduce the performance of Wan2.1-14B and Wan2.2-A14B (Wang et al., 2025) on the T2V-CompBench to establish strong baselines. All models generate videos under their optimal inference settings. To ensure a fair comparison between the two representative compositional generation paradigms, LVD (Lian et al., 2024) and Vico (Yang & Wang, 2024), we reproduce them along with Comp-Attn on the same video foundation models for comparison. Both methods utilize gradients computed from the cross-attention map for video latent optimization during the testing phase, while LVD explicitly uses LLM-planned layouts for supervision. LVD uses the same layout as Comp-Attn. All hyperparameters are set according to their optimal configurations in the original papers.

As for additional T2I task evaluation, we use FLUX (flu, 2024) along with the results reported for HybridLayout (Wu et al., 2025) and SiamLayout (Zhang et al., 2025a) on T2I-CompBench (Huang et al., 2025) for a fair comparison. We also test the performance of applying Comp-Attn to FLUX on T2I-CompBench using the same prompt suite and evaluation process.

## C. More Ablation Results on SCI

Recall that SCI performs subject token enhancement by first conducting subject saliency estimation to obtain significance scores for each subject, followed by interpolation computation. Therefore, we aim to investigate whether directly interpolating subject anchor embeddings into the prompt at certain ratios could also be effective. Therefore, we establish a baseline formulated as follows:

$$\boldsymbol{P}_i^* = (1 - \alpha)\boldsymbol{P}_i \oplus \alpha \overline{\boldsymbol{A}}_i. \tag{13}$$

*Table 8.* Ablation study on the subject saliency estimation in SCI.

| Method | Con-attr | Spatial | Interact | Action |
|---|---|---|---|---|
| w/ Interpolation $\alpha = 0.2$ | 0.8426 | 0.6694 | 0.8065 | 0.7839 |
| w/ Interpolation $\alpha = 0.5$ | 0.8331 | 0.6785 | 0.8206 | 0.7924 |
| w/ Interpolation $\alpha = 1.0$ | 0.8348 | 0.6650 | 0.7981 | 0.7802 |
| **SCI (Ours)** | **0.8613** | **0.6840** | **0.8357** | **0.8022** |

*Figure 6.* **More qualitative comparisons illustrating the impact of Comp-Attn on FLUX.1-dev.**

where $\alpha$ indicates the mixing rates, which is a hyperparameter. As demonstrated in Table 8, directly interpolating subject anchor embeddings into prompts with fixed mixing rates yields only marginal improvements, significantly inferior to SCI. This performance gap primarily stems from the dynamic nature of subject saliency across varying prompt contexts. Hence, manually defined mixing parameters fails to adapt to these variations, potentially resulting in either insufficient restoration of original subject semantics or corruption of contextual information stored in subject tokens.

## D. More details for LLM Planning Layout

We utilize GPT-4o-mini (GLM et al., 2024) to generate coarse layouts for each prompt and extract the specified subjects. Invoking the layout planning API for video generation incurs a cost of approximately $0.005 per request with a response latency around five seconds. Both the qualitative and quantitative results in this study are based on the GPT-4o-mini planned layout to achieve efficient batch experiments. Notably, users can also customize object layouts to enable diverse compositional generation. For detailed generation prompts and the corresponding data format, please refer to Figure 13.

## E. More Qualitative Results

We present extended qualitative results demonstrating Comp-Attn's consistent superiority when adapted to other video generation architectures, Wan2.2-A14B (Figure 7, Figure 8, Figure 9), Wan2.1-14B (Figure 10), CogVideoX (Figure 11) as well as VideoCrafter2 (Figure 12). Comp-Attn achieves precise multi-subject presence while ensuring accurate alignment of inter-subject relations. Moreover, Comp-Attn can be robustly extended to the image generation frameworks such as FLUX, further enhancing the compositional semantic understanding of T2I models (Figure 6).

## F. Robustness of LLM layout.

This section first analyzes the robustness and error rate of using LLM directly for layout planning. Then, we elaborate on the two most common types of errors, followed by a discussion on how our method effectively addresses these issues. We systematically analyzed the layouts planned by GPT-4o-mini for 1,400 prompts in the T2V-CompBench dataset, manually

*Table 9.* Layout error comparison under different settings. "Score" represents the average score of T2V-CompBench.

| Method | Size error ↓ | Motion error ↓ | Error rate ↓ | Score ↑ |
|--------|--------------|----------------|--------------|---------|
| Valina | 19 | 22 | 2.93% | 0.6358 |
| + Ours | 11 | 5 | 1.14% | 0.6414 |

verified their validity, and calculated the overall error rate.

A direct paradigm for generating layouts using LLMs could be: prompting the LLM to generate a layout for each latent frame, using an in-context example as a demonstration. However, under this dense planning setup, the model often struggles to balance the spatial relationships across too many frames and the transitions between them, leading to conservative motion planning in terms of temporal dynamics and incorrect size planning in terms of spatial arrangements. We categorize the errors under this setup into: 1) motion error and 2) size error. Specifically, motion error is typically manifested as the layout planned for an object remaining static or exhibiting minimal movement across multiple frames (e.g., the total movement distance being only 0.1 times the video width). Meanwhile, size error is mainly reflected in certain objects in the video being assigned an unreasonably small or disproportionately sized bounding box (e.g., with a total area accounting for only 1% of each frame). As shown in Table 9, we found that under the dense planning setup, the error rate of the layout is 2.93%. While this is not particularly significant, it does have an impact on certain performances.

To address this issue, we adopt a strategy of ***planning keyframes while interpolating the redundant frames***. The motivation primarily stems from two key observations: 1) simplifying the planning task and requiring the LLM to generate layouts for fewer frames improves its understanding of object spatial scales, while the motion between keyframes becomes relatively more reasonable; 2) under the dense planning setup, the layouts of many frames exhibit a linear and simple pattern of relative motion, resulting in significant redundancy. As shown in Figure 13, based on this observation, we first let the LLM plan 4 keyframes to determine the object's size and approximate motion trajectory. Then, we efficiently generate the intermediate frames using linear interpolation. Ultimately, our method significantly reduces the layout error rate, further enhancing the robustness of LLM-planned layouts. At the same time, the design mechanism of Comp-Attn can tolerate a certain degree of size error. For instance, when the bounding box of an object is too small, the enhanced semantics of subject tokens through SCI allow the object to still be recalled. Additionally, the dynamic attention modulation mechanism of LAM can center on this small bounding box and roughly present the object at the corresponding location. This contrasts with previous methods (Tian et al., 2024; Wang et al., 2026) that relied overly on LLM-planned layouts and adopted hard mask mechanisms.

Overall, the LLM-planned layout is relatively robust, with an initial error rate of 2.93% that our strategy successfully reduces to 1.14%. The prompt strategy we adopted, combined with the inherent robustness of Comp-Attn, further minimizes the impact of errors in the planned layout. Future improvements could involve training a dedicated LLM for text-to-layout tasks or adopting a human-in-the-loop approach for layout planning.

## G. Failure case analysis.

Comp-Attn may still fail when handling prompts that require rich physical knowledge. As shown in Figure 14, when a prompt contains multiple elements and involves complex physical laws, the baseline model generates videos with compositional semantic errors and misalignment with physical laws. Although Comp-Attn improves the presentation of compositional semantics, it still produces content that violates common physical sense. This is mainly due to the lack of physical world knowledge in current foundational video generation models.

## H. Limitations.

While Comp-Attn significantly improves composition awareness in video generation, its performance is still limited by the inherent constraints of the underlying foundation models. For example, when the foundation model lacks sufficient physical knowledge, it may also generate lower-quality content. Despite being model-agnostic by design, the effectiveness of Comp-Attn heavily relies on the semantic grounding and representational capabilities of the pretrained backbone model. Future research could focus on enhancing the reasoning ability for multi-subject representation and inter-subject relationships through targeted training, further driving advancements in the field of compositional generation.

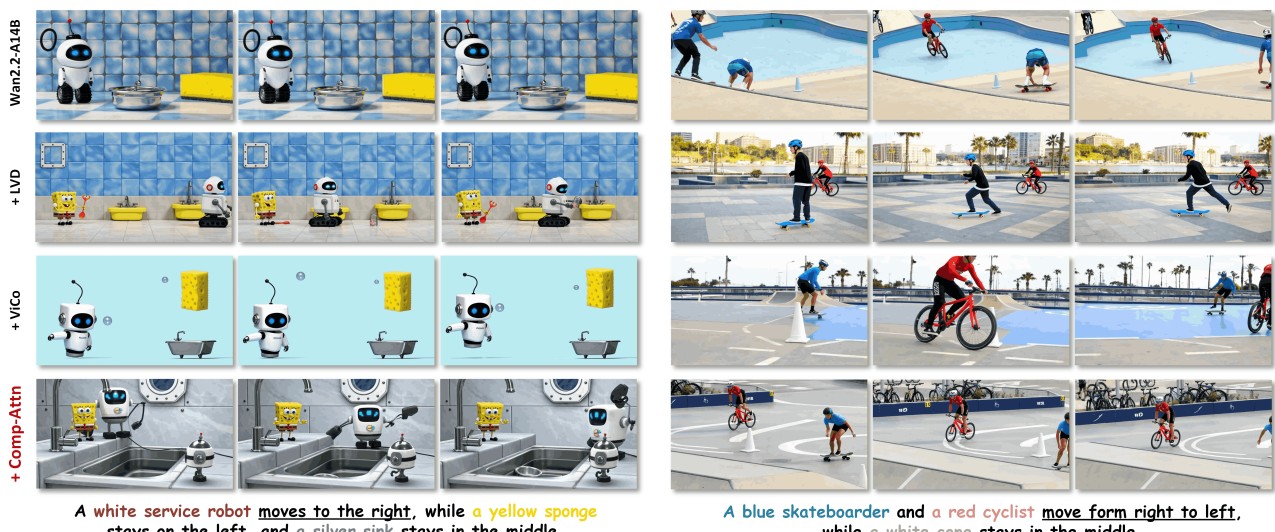

*Figure 7.* **More qualitative comparisons illustrating the impact of Comp-Attn on Wan2.2-A14B.**

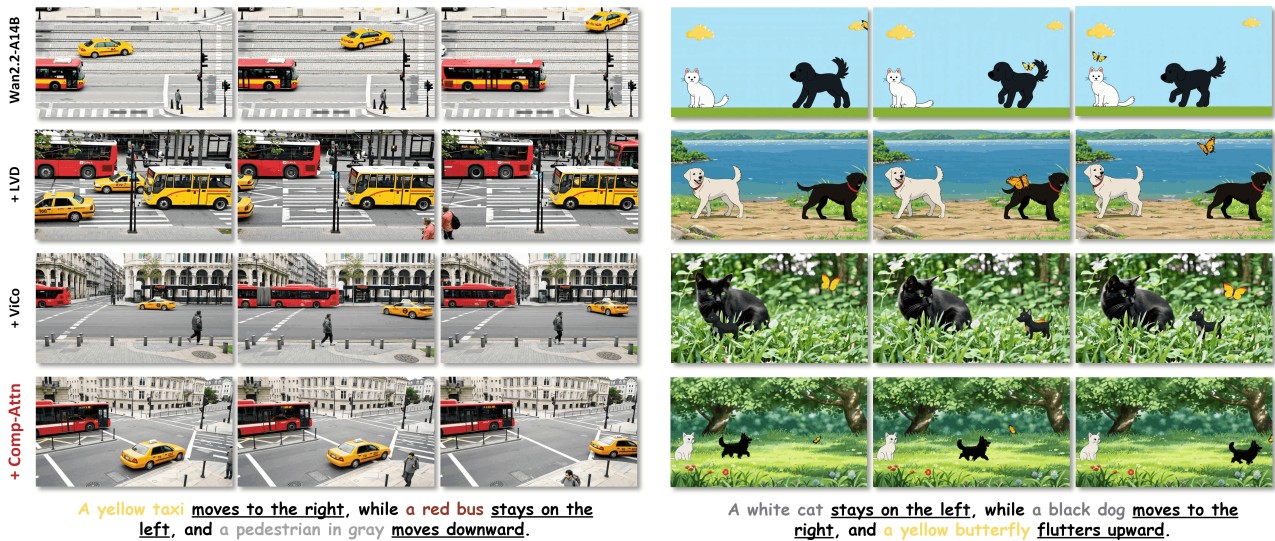

*Figure 8.* **More qualitative comparisons illustrating the impact of Comp-Attn on Wan2.2-A14B.**

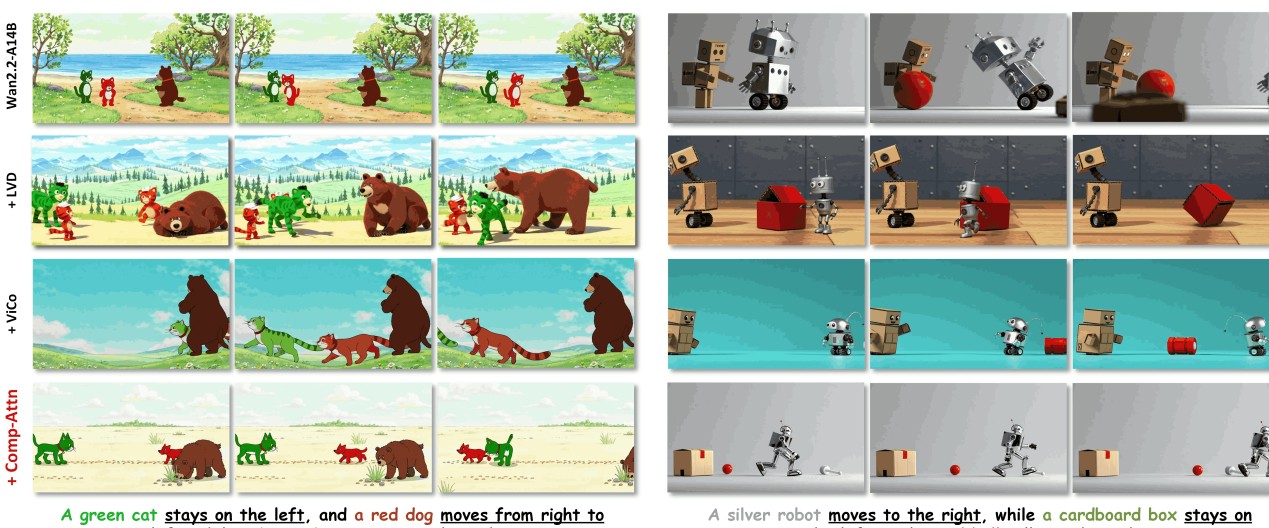

*Figure 9.* **More qualitative comparisons illustrating the impact of Comp-Attn on Wan2.2-A14B.**

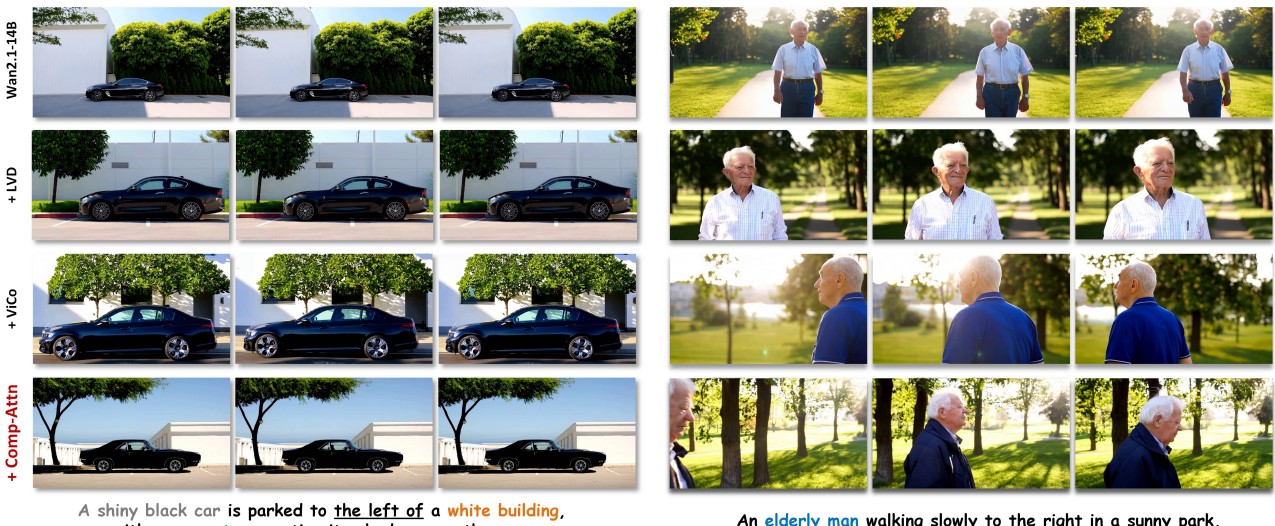

*Figure 10.* **More qualitative comparisons illustrating the impact of Comp-Attn on Wan2.1-14B.**

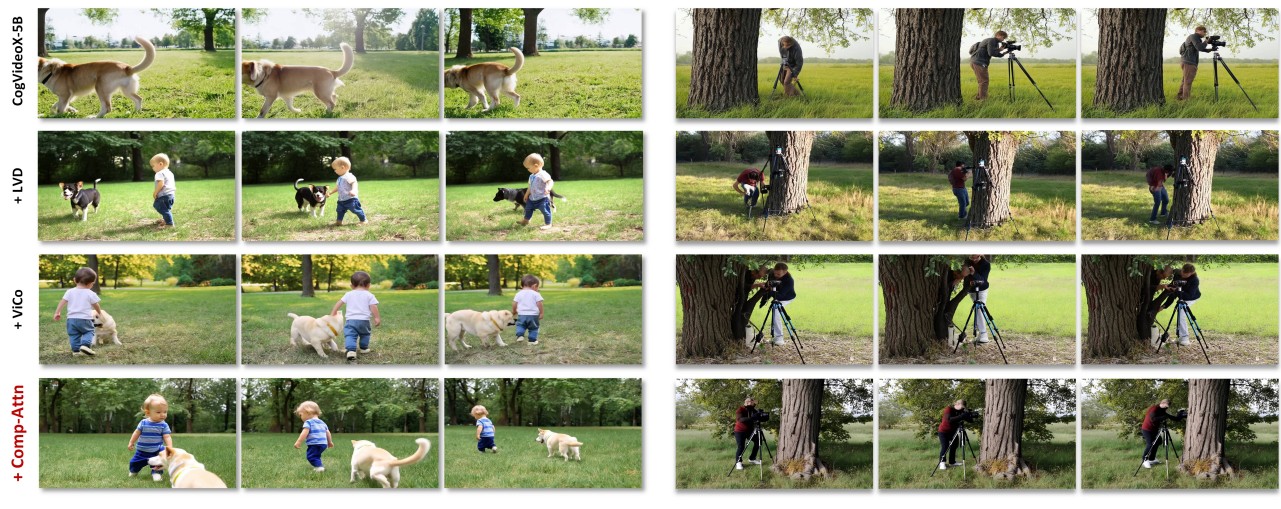

*Figure 11.* **More qualitative comparisons illustrating the impact of Comp-Attn on CogVideoX-5B.**

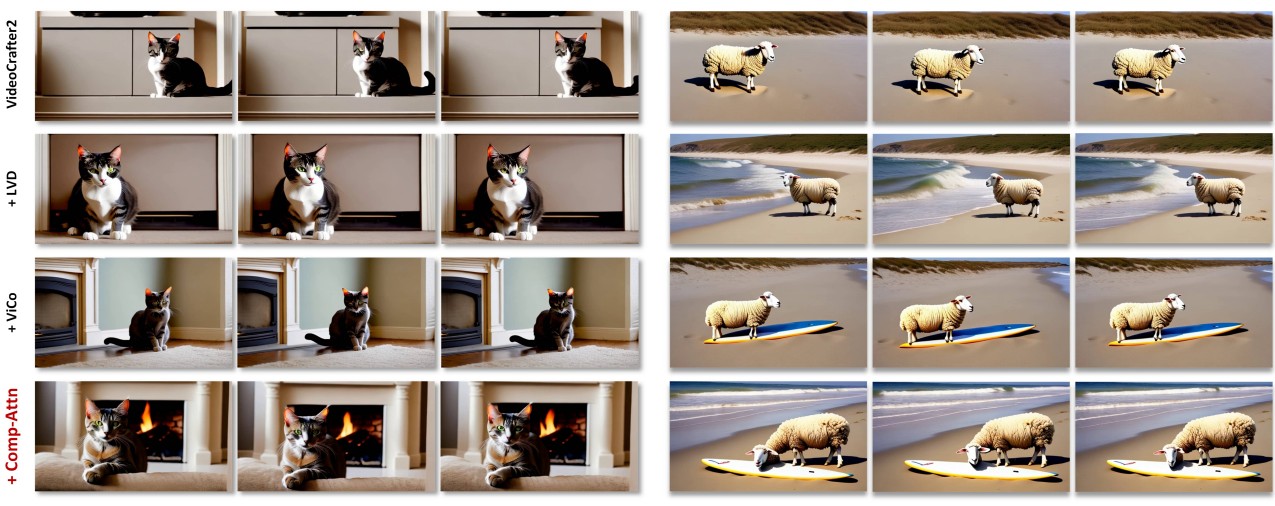

*Figure 12.* **More qualitative comparisons illustrating the impact of Comp-Attn on VideoCrafter2.**

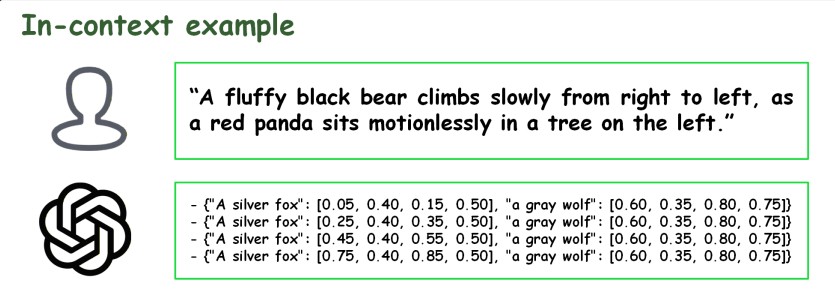

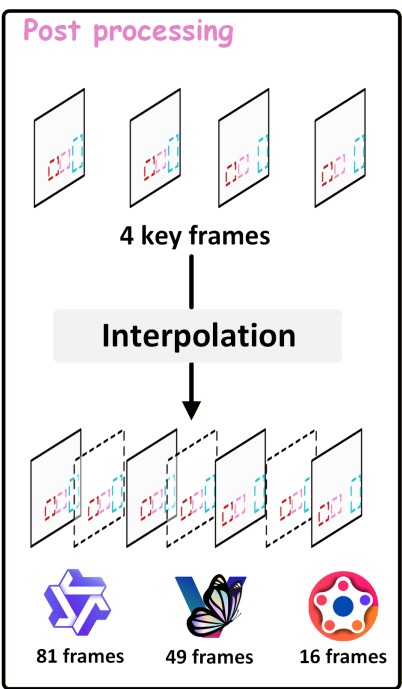

*Figure 13.* **Prompt template for LLM layout generation.** We first let the LLM plan 4 keyframes to indicate the precise size of objects and their key positions in the video. Subsequently, we use an efficient linear interpolation method to obtain the layouts for all frames.

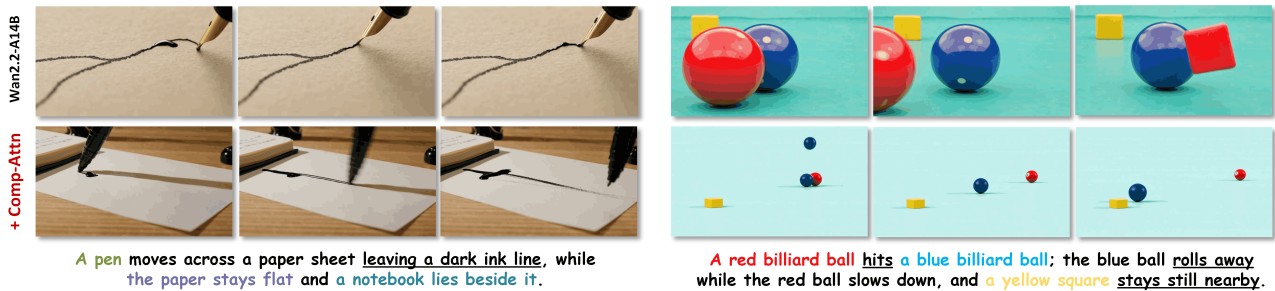

A pen moves across a paper sheet leaving a dark ink line, while the paper stays flat and a notebook lies beside it.

A red billiard ball hits a blue billiard ball; the blue ball rolls away while the red ball slows down, and a yellow square stays still nearby.

*Figure 14.* **Failure case analysis.** When handling prompts that require rich physical knowledge, although Comp-Attn improves compositional semantics, it can still generate motions or phenomena that are misaligned with physical laws (e.g., unnatural liquid flow, interpenetrating collisions, etc.).

