# OpenReview forum: "Comp-Attn: Present-and-Align Attention for Compositional Video Generation"
_ICML.cc/2026/Conference — ICML 2026 regular_

### Official Review · Reviewer_6Cgj · 2026-02-25

**Soundness:** 3
**Presentation:** 3
**Significance:** 2
**Originality:** 3
**Overall Recommendation:** 4
**Confidence:** 5

**Summary:**

This paper introduces Comp-Attn, a compositional video generation framework that aims to address two primary failure modes in this topic: subject absence and misaligned relations. To solve this, the authors propose a training-free and plug-and-play cross-attention mechanism that follows a "Present-and-Align" paradigm. Specifically, the paradigm includes two components:
- Subject-aware condition interpolation (SCI) which reinforces subject-specific text conditioning to improve subject presence
- Layout-forcing attention modulation (LAM) which aligns spatial relations by dynamically adjusting cross-attention distributions using IoU guidance based on LLM-planned layouts.

The authors verify Comp-Attn on four video generation backbones Wan2.1, Wan2.2 (DiT), CogVideoX (MM-DiT), and VideoCrafter2 (U-Net). The results on T2V-CompBench and T2I benchmarks show significant performance gains with small inference overhead.

**Compliance With Llm Reviewing Policy:**

Affirmed.

**Final Justification:**

I have carefully read the authors' latest response on April 3. Here is my final comment and justification:

**Fully-resolved concerns**
- [W1] Flash-attention compatibility
- [W4] Heuristic hyperparameter sensitivity

**Partially-resolved concerns**
- [W2] Direct evaluation on "presence vs. relation": the authors provide a new human and GPT-5 evaluation on 200 sampled prompts, which is a better and more direct validation on "Present-and-Align" motivation. However, for the prompts that Comp-Attn still fails, what is the proportion that can be attributed to subject absence v.s. relational misalignment? That is to say, without these results, it is hard to clarify whether these two failure modes are independent.
- [W3] Layout perturbation robustness: thanks for the authors for providing the anonymous link, however, I am unable to access it and therefore cannot verify this claim (not sure whether it is a technical issue). I encourage the authors to ensure these examples are included in the revised paper.

Overall, this paper presents a well-motivated / training-free / plug-and-play approach to compositional video generation. The rebuttal has strengthened the paper, while a few minor concerns (e.g., failure-case analysis and the depth of the presence/relation decomposition) are shared across multiple reviewers.

I maintain my rating of weak accept, and highly encourage the authors to include all of the content provided in the rebuttal into the paper, which can largely strengthen the paper.

**Key Questions For Authors:**

- [Q1] Could Comp-Attn be compatible with Flash Attention? If no, I hypothesize there is a huge degradation on the inference speed.
- [Q2] Could the authors discuss the evaluation of "presence vs relation" (either through the provided experiments or through new experiments)? It could better strengthen the verification of the main statement/claim in the paper.
- [Q3] Could the authors provide some failure cases when there are systematic layout perturbations from LLM?

**Limitations:**

Yes, the authors discuss the limitations of the proposed method in Appendix H, and provide an impact statement after the Conclusion section.

**Strengths And Weaknesses:**

**Strengths**
- [S1] The proposed method is well-motivated. The authors first identity two common failure cases (object presence + relations) then correspondingly designs a decoupled cross-attention paradigm, which sounds reasonable.
- [S2] I like the idea of IoU-guided LAM which applied the soft attention modulation based on the layouts from LLM. It allows flexibility in subject shapes and maintains better visual quality while ensuring spatial accuracy in comparison of hard attention.
- [S3] I also like the experiments regarding the comparisons of inference latency, as using LLM to plan layouts could a huge bottleneck of inference efficiency. With such results, my concerns on the computation overhead has been addressed.
- [S4] The proposed method is a plug-and-play module and the authors provide results on a wide range of video generation backbones (DiT/MM-DiT/U-Net). The setting itself and the experiments are sufficiently convincing that Comp-Attn be effective in a lot of usage scenarios.
- [S5] The authors provide a well-prepared supplementary material which includes the source code and original video files of all the results shown in the paper. It can largely improve the reproducibility of the proposed method and also makes the reviewers better justify the model performance.

**Weaknesses**
- [W1] My major concern is that most of the current DiTs use Flash Attention to accelerate the attention computation, based on the assumption of "regular tiling". However, the proposed LAM generate arbitrary-shaped masks and does not follow this hypothesis. Does Comp-Attn use Flash Attention? How do the authors solve this issue?
- [W2] I cannot find a direct evaluation on "presence vs relation" (which is the main motivation of the paper). From my understanding, there are no metrics in T2V-CompBench which clearly map to these two issues.
- [W3] The method uses GPT-4o-mini for subject parsing/layout planning, which however often generates hallucinations. Although the authors extensively discuss the robustness of LLM layout in Appendix F, I believe it could be more complete if the authors also provide some sensitivity studies: for example, how compositional performance changes under systematic layout perturbations, (probably provide some failure cases under these circumstances).
- [W4] Some components are hand-crafted and rely on heuristic design choices. For example, 1) SCI has a fixed temperature of 0.2 and uses a linear timestep attenuation, and 2) LAM thresholds attention maps by mean to derive an adaptive-perception layout. Again, while the authors provide ablations on some key designs, I am still concerns with the model stability across different prompt distributions/subject counts/some challenging deformation regimes.

**Suggestion**
- [MW1] There is a typo in the submission title: "Genneration" should be "Generation".

---

> ### Author Rebuttal · Authors · 2026-03-31
>
> We appreciate the time and effort you put into reviewing our submission. Thanks for recognizing the motivation behind our method and its overall design. We will revise the typo issue. We now address your concerns below:
>
> > ### **W.1 & Q1: Flash attention compatibility**
>
> We agree that introducing an additional logit bias in cross-attention can prevent using  Flash Attention kernel in our current implementation. However, Comp-Attn is applied **only to cross-attention** and **only during the first 20% denoising steps**, while the overall compute of a DiT block **is dominated by self-attention**.
>
> Concretely, for an 81×832×480 video, the self-attention matrix is approximately (21×52×30)×(21×52×30), whereas cross-attention is only (21×52×30)×256, yielding roughly a **128:1** ratio in attention-matrix size. Therefore, we keep Flash Attention for all self-attention (no mask/bias applied), and for cross-attention we currently switch to a PyTorch backend only at the timesteps where Comp-Attn is enabled (to apply the bias term).
>
> With this setup, the additional runtime remains controlled: as shown in **Table A**, we incur only a **+3.9%** inference overhead compared to the baseline where both self-/cross-attention use Flash Attention.
>
> **Table A: Inference latency comparison on a single H100 GPU. LLM API latency is included in Comp-Attn.**
> | Model        | Wan2.2-A14B | + LAM & SCI| + Comp-Attn (full) |
> |--|--:|--:|--:|
> | Latency (s) | 411 | 427 | 432 |
>
> Finally, this overhead is primarily an engineering issue rather than an algorithmic limitation. In future revisions, it is feasible to further reduce it by implementing a custom attention kernel that supports our bias form for cross-attention.
>
>
> > ### **W.2 & Q2: Direct evaluation on "present and align"**
>
> Although T2V-CompBench do not explicitly state the presence and relational alignment, the evaluation pipelines of most metrics **implicitly include both**. For example, in **Grid-LLaVA** for **action** and **interaction**, it first checks whether the subjects are present and then evaluates whether the relations and interaction bindings between them are correct. In the **spatial** metric, G-Dino first verifies that each subject exists and then computes whether the specified spatial relationship is correct. Therefore, these metrics essentially test the **“presence and relational alignment”** emphasized in our motivation.
>
> > ### **W3 & Q3. Robustness to layout perturbations**
>
> Thanks for the suggestion. We test layout sensitivity on the key metrics of T2V-CompBench by perturbing boxes with random scaling (0.5×–1.5×) and randomly freezing box updates across adjacent frames. **Table B shows** only slight drops, suggesting robustness to moderate layout noise. This is largely because **LAM performs soft attention modulation**, treating the layout as a soft prior rather than a hard constraint, and thus tolerates moderate layout errors.
>
> **Table B: Layout sensitivity analysis.**
> | Method | Con-Attr | Spatial | Motion |
> |---|---:|---:|---:|
> | Wan2.2-A14B | 0.8394 | 0.6618 | 0.3654 |
> | Comp-Attn | 0.8726 | 0.7244 | 0.4708 |
> | Comp-Attn (w/ noise) | 0.8681 | 0.7162 | 0.4629 |
>
> We will also include failure cases under extreme, manually perturbed layouts in the revised version, e.g., overly small boxes that make subjects unresponsive, or fully static layouts across the whole video that hinder object motion. However, such extreme errors are rare in practice (Appendix F).
>
> > ### **W.4: Stability to hyperparameters**
>
> Thanks for the question. Although Comp-Attn include some heuristic choices, we find the method is not highly sensitive to these settings.
>
> For SCI, we add a temperature sensitivity ablation: varying τ only causes minor performance changes (**Table C**). This is expected since τ mainly affects the sharpness of saliency weights rather than the semantic injection direction. We also ablate the linear decay w(t): removing it leads to a moderate drop (**Table D**), consistent with our claim that stronger early-stage injection helps subject presence, while the main gain comes from SCI itself.
>
> **Table C: More ablation results on τ.**
> | τ | 0.1 | 0.2 | 0.5 | 1.0 |
> |--:|--:|--:|--:|--:|
> | T2V-CompBench (overall) |  0.6378|  0.6414|  0.6386|  0.6325|
>
>
> **Table D: More ablation results on w(t).**
> |  | w/o w(t)| Ours|
> |--:|--:|--:|
> | T2V-CompBench (Avg) |  0.6291|  0.6414|
>
> For LAM, thresholding by the mean of the attention map is a simple layout-extraction rule [1][2] that adapts across prompts without introducing hyperparameters.
>
> Overall, Comp-Attn shows low sensitivity to these hyperparameters and remains stable across diverse prompt configurations, as reflected by its consistent results over the 1,400 prompts in T2V-CompBench.
>
> [1] ConceptAttention: Diffusion Transformers Learn Highly Interpretable Features, ICML 2025.
>
> [2] DiTCtrl: Exploring Attention Control in Multi-Modal Diffusion Transformer for Tuning-Free Multi-Prompt Longer Video Generation, CVPR 2025.

---

> > ### Author Rebuttal · Reviewer_6Cgj · 2026-04-03
> >
> > I appreciate the authors providing detailed responses and new experiments. I have carefully read them as well as the comments from other reviewers. Here are my feedbacks:
> >
> > **[W1] Flash attention compatibility**
> > - The authors clarify that Comp-Attn only applies during the first 20% of denoising steps, and the cross-attention matrix is ~128× smaller than self-attention.
> > - Table A shows that Comp-Attn only has only 3.9% computation time overhead, which is negligible.
> > - My concern has been fully resolved.
> >
> > **[W2] Evaluation on "presence v.s. relation"**
> > - The authors argue that T2V-CompBench metrics capture both aspects (e.g., Grid-LLaVA checks subject presence before evaluating relational correctness).
> > - But I still think the paper should have a "breakdown". For example, you can show a per-prompt analysis and see how often failures are due to missing subjects vs. incorrect relations. This would more directly validate the main motivation of the paper (i.e., "Present-and-Align").
> > - A follow-up question: could you provide a brief test (even on a subset of T2V-CompBench) to show the proportion of failure cases attributable to subject absence vs. relational misalignment, with and without Comp-Attn? This could more directly support the paper's motivation and help readers understand where each component (SCI v.s. LAM) contributes most.
> > - My concern is partially addressed.
> >
> > **[W3] Sensitivity about layout perturbation**
> > - Table B shows that layout perturbation only causes slight degradation.
> > - The authors also promise that they will include failure cases under extreme perturbations in the revision.
> > - My concern is partially addressed (not sure how the failure cases will be like).
> >
> > **[W4] Heuristic hyperparameters**
> > - Tables C/D show new ablations and show low sensitivity across different hyperparameter selections.
> > - I also appreciate the authors provide the reference to prior work for mean-thresholding in LAM.
> > - My concern is fully addressed.
> >
> > **Shared concerns across reviewers**
> > - I also noticed that several of my concerns were mentioned by other reviewers.
> > - Similar to my [W3], Reviewers fPPr/bzau also mentioned the robustness and reliability of LLM-based layout planning.
> > - Similar to my [W4], Reviewer bzau also mentioned the sensitivity issue of the hyperparameter.
> >
> > Overall, the rebuttal has addressed some of my concerns. The quality of the paper is generally good, but also with some noticeable minors/concerns shared across multiple reviewers. I maintain my rating as weak accept and am open to further discussion with authors/other reviewers/ACs.

---

> > > ### Author Response · Authors · 2026-04-04
> > >
> > > We sincerely thank you for your detailed feedback and constructive suggestions, which have helped improve our work. We are glad that W1 and W4 have been satisfactorily addressed, and we will now respond in detail to the remaining issues for W2 and W3.
> > >
> > > > ### **W.2: Evaluation on "presence v.s. relation"**
> > >
> > > We agree that directly and quantitatively validating “Presence” and “Align” would better support our main motivation. To this end, we sampled 200 prompts from T2V-CompBench to validate this. Here, “Presence” fails if any required subject is missing or incorrect, and “Relation alignment” fails if the prompted inter-subject relation is violated. Our evaluation consists of two parts: 1.We recruited three volunteers to manually assess each video for subject presence and relation alignment. Using a binary (0/1) decision, we computed the failure rate for each criterion and reported the average results. 2.We additionally prompted GPT-5 to perform an extra round of failure-rate assessment. The detailed experimental results are shown in **Table A** and **Table B** below:
> > >
> > > **Table A: Direct evaluation on “Presence” (failure rate ↓).**
> > > | Evaluator | Wan2.2-A14B | +SCI | +LAM | +LAM & SCI (Comp-Attn) |
> > > |---|---:|---:|---:|---:|
> > > | Human |  34.5% |  22.0% |  30.0% | 18.0% |
> > > | GPT-5 |  32.5% |  20.0% |  28.5% | 15.5% |
> > >
> > > **Table B: Direct evaluation on “Relation alignment” (failure rate ↓).**
> > > | Evaluator | Wan2.2-A14B | +SCI | +LAM | +LAM & SCI (Comp-Attn) |
> > > |---|---:|---:|---:|---:|
> > > | Human | 39.5% | 34.5% | 26.0% | 22.5% |
> > > | GPT-5 | 35.0% | 32.0% | 24.0% | 19.5% |
> > >
> > > **SCI** mainly improves “Presence”, while **LAM** mainly improves “Relation alignment”, which aligns with our motivation. Meanwhile, they also bring complementary gains to each other to some extent: SCI increases subject presence, serving as a prerequisite for relation alignment; LAM’s attention modulation strengthens interactions between some pixels and the subject text tokens, thereby slightly improving subject rendering even without SCI refinement. Overall, combining them yields the best performance gains.
> > >
> > > > ### **W.3: Sensitivity about layout perturbation**
> > >
> > > Thanks for the suggestion. We have now added these failure cases under extreme perturbed layouts (e.g., overly small boxes that prevent subjects from responding within the desired regions, or fully static layouts across the whole video that hinder object motion). Notably, such errors are rare in practice, as stated in Appendix F. The concrete examples are provided in our anonymous link:
> > >
> > > https://anonymous.4open.science/r/Comp-attn-4D02/R3_W3.md
> > >
> > > We hope these additions address your remaining concerns on W2 and W3. If you feel the main issues are resolved, we would also be grateful if you could consider whether this affects your overall assessment or score.

---

### Official Review · Reviewer_fPPr · 2026-03-12

**Soundness:** 3
**Presentation:** 3
**Significance:** 3
**Originality:** 2
**Overall Recommendation:** 4
**Confidence:** 2

**Summary:**

Comp-Attn addresses subject presence and inter-subject relations in compositional video generation by employing Subject-aware Condition Interpolation (SCI) to reinforce subject-specific semantics, and Layout-forcing Attention Modulation (LAM) to synchronize attention distributions with planned relational layouts.

**Compliance With Llm Reviewing Policy:**

Affirmed.

**Key Questions For Authors:**

I am wondering whether it is appropriate for a ‘training-free’ approach to rely on an external LLM for spatial reasoning.

**Limitations:**

See Weaknesses

**Strengths And Weaknesses:**

Strengths:
1. This work is clearly expressed and easy to understand.
2. The method is training-free and can be directly applied during inference.
3. The paper resolves the issues of subject omission and misaligned spatial relationships.

Weaknesses:
1. The motion states demonstrated in the paper’s examples appear relatively simplistic, primarily involving linear trajectories such as left-to-right movement. Does the framework’s capacity for synthesizing complex, non-linear motion remain constrained by the strategy of utilizing sparse LLM-generated keyframes followed by linear interpolation?
2. Furthermore, when a significant discrepancy arises between the external LLM’s spatial plan and the foundation model’s inherent distribution, can the generation still effectively adhere to the provided layout? It is critical to consider whether such a mismatch might induce distorted or unnatural motion artifacts, rather than achieving a faithful alignment with the intended plan.

---

> ### Author Rebuttal · Authors · 2026-03-31
>
> Thank you for your careful review and constructive feedback. We appreciate your positive assessment of our motivation and overall approach. We address your concerns below:
>
> > ### **W.1: Complex and non-linear motion synthesis**
>
> Thanks for the question. Keyframe-based planning with interpolation is our default efficient layout-generation protocol to enable systematic benchmarking. It is a practical choice for evaluation and does not reflect an intrinsic limitation of our method. Importantly, Comp-Attn does not assume linear trajectories. It modulates attention **frame-by-frame** according to the provided layouts, so it can follow non-linear motions as long as the layout sequence specifies them (e.g., via denser keyframes or user-specified frame-wise layouts). In fact, users can directly customize each subject’s trajectory to enable more diverse and personalized motions beyond simple left-to-right movement. We additionally provide an anonymous demo link **[Link1]** with **circular and out-and-back** user-specified trajectories, where Comp-Attn consistently preserves all subjects and the realized motions closely match the intended layouts.
>
> [Link1] https://anonymous.4open.science/r/Comp-attn-4D02/R2_W1.md
>
> > ### **W.2: Distribution mismatch between LLM layouts and the foundation model**
>
> Thanks for the question. In our experiments, we do not observe a systematic distribution mismatch between LLM-planned layouts and the video foundation model priors. As discussed in the Appendix, most LLM-planned layouts are reasonable in our setting and largely match the video foundation model’s priors on object scale and motion magnitude. Accordingly, noticeable planning errors are rare (only ~1.1% prompts show clear motion/size issues; see Appendix F).
>
> Moreover, **Comp-Attn uses IoU-guided soft attention modulation, which treats the planned layout as a reference rather than a hard condition**. This design encourages the generated subjects to align with the intended spatial regions while still allowing the foundation model to rely on its learned priors, thereby reducing the risk of distorted or unnatural motion when mismatches exist. Moreover, **Table A** shows consistent gains with IoU-guided soft modulation, indicating better layout alignment while preserving natural motion under plan–prior mismatch.
>
> **Table A: Ablation on IoU-guided soft modulation.**
> | Method             | Con-attr | Spatial | Interact | Action |
> |--------------------|---------:|--------:|---------:|-------:|
> | w/o IoU-guided soft modulation  | 0.8364   | 0.6928  | 0.7981   | 0.7850 |
> | LAM (Ours)         | 0.8595   | 0.7079  | 0.8296   | 0.8213 |
>
> We further conduct a layout robustness study on T2V-CompBench by injecting perturbations to the box layouts, including random scaling (0.5×–1.5×) and randomly freezing box updates for adjacent frames to simulate planning noise and plan–prior mismatch. As shown in **Table B**, performance drops only slightly under these perturbations, indicating low sensitivity to moderate layout errors.
>
> **Table B: Layout sensitivity analysis.**
> | Method | Con-Attr | Spatial | Motion |
> |---|---:|---:|---:|
> | Wan2.2-A14B | 0.8394 | 0.6618 | 0.3654 |
> | Comp-Attn | 0.8726 | 0.7244 | 0.4708 |
> | Comp-Attn (w/ noise) | 0.8681 | 0.7162 | 0.4629 |
>
> ### **Q.1: Using LLM-based layout planning in a training-free framework**
>
> Thanks for the question. In our paper, “training-free” means that we perform no additional training or fine-tuning in the pipeline. That's to say, no backpropagation and no parameter updates to the video foundation model and other tools. The external LLM is only used as an inference-time planner to translate text into a structured layout sequence.  Importantly, our method only requires a layout sequence as input, which can be provided through multiple sources (e.g., user-specified trajectories and boxes, dataset annotations, rule-based planners, or other tools).
>
> We adopt an LLM planner simply as a convenient and effective option. The LLM provides only high-level spatial plans, while our main contribution lies in how to inject layouts into the foundation model via IoU-guided soft attention modulation. Therefore, using an external LLM for spatial reasoning is fully compatible with the training-free setting. This is also consistent with prior “training-free” video generation methods [1][2][3] that use LLMs purely at inference time, e.g., as prompt rewriters or reasoning modules, without any parameter updates.
>
> [1] MEVG: Multi-event Video Generation with Text-to-Video Models, ECCV 2024.
>
> [2] DiTCtrl: Exploring Attention Control in Multi-Modal Diffusion Transformer for Tuning-Free Multi-Prompt Longer Video Generation, CVPR 2025.
>
> [3] Chain of Event-Centric Causal Thought for Physically Plausible Video Generation, CVPR 2026.

---

### Official Review · Reviewer_bzau · 2026-03-13

**Soundness:** 3
**Presentation:** 3
**Significance:** 3
**Originality:** 3
**Overall Recommendation:** 4
**Confidence:** 3

**Summary:**

This paper proposes Comp-Attn, a training-free cross-attention mechanism designed to improve compositional text-to-video (T2V) generation. The authors identify two primary challenges in generating compositional scenes: subject presence (omission of entities) and inter-subject relations (misaligned spatial or interactive relationships). To address these, the framework introduces a "Present-and-Align" paradigm. The first component, Subject-aware Condition Interpolation (SCI), reinforces individual subject semantics during the text conditioning phase to ensure all subjects are present. The second component, Layout-forcing Attention Modulation (LAM), softly aligns the video-text attention distributions with LLM-generated bounding box layouts using dynamic Intersection over Union (IoU) guidance. Experimental results demonstrate that Comp-Attn can be integrated as a plug-and-play module into various diffusion models (such as Wan2.2, CogVideoX, and VideoCrafter2) , yielding significant quantitative improvements on the T2V-CompBench dataset with minimal inference latency overhead.

**Compliance With Llm Reviewing Policy:**

Affirmed.

**Key Questions For Authors:**

1. How sensitive is the final video quality to inaccuracies in the initial LLM-planned layouts, and have you considered using smaller, locally hosted visual language models to mitigate the 5-second API latency overhead?
2. Can you provide empirical justification or ablation studies for the chosen hyperparameters, specifically the temperature factor set to 0.2 and the linear decay schedule?

**Limitations:**

See above

**Strengths And Weaknesses:**

Strengths:

1. The proposed Comp-Attn module is highly versatile, as it can be seamlessly integrated into existing text-to-video diffusion models across different architectures (like Wan2.2, CogVideoX, and VideoCrafter2) without requiring any further training.

2. The "Present-and-Align" paradigm is intuitively designed, effectively decoupling the compositional generation problem by handling subject presence at the condition level through SCI and relational alignment at the attention-distribution level via LAM.

3. Unlike prior layout-control methods that enforce rigid bounding box constraints, the LAM module utilizes an IoU-guided soft attention modulation strategy. This flexible feedback mechanism successfully balances original attention with layout-region attention, avoiding the excessive modulation that often degrades visual diversity.

Weakness:
1. The LAM module's reliance on a large language model (GPT-4o-mini) to generate prior bounding box layouts introduces a noticeable bottleneck. The approximately 5-second latency per API request could significantly limit the framework's applicability in real-time or high-throughput scenarios.

2.The method struggles with complex physical interactions, which the authors acknowledge in their failure cases.  Since the framework manipulates spatial attention without injecting actual physical priors, it frequently generates phenomena that violate basic physical laws, such as unnatural liquid flow or interpenetrating collisions.

3.The methodology relies on several fixed hyperparameter choices, such as setting the temperature factor $\tau$ to 0.2 for Subject Saliency Estimation. The manuscript currently lacks a comprehensive sensitivity analysis to demonstrate how robust the framework is to variations in these parameters.

4. The quantitative evaluation primarily benchmarks against foundation models augmented with other inference-time optimization techniques, such as LVD and ViCo. The paper would be much stronger if it included a discussion or empirical comparison with models that natively train on spatial layout controls, providing a clearer picture of where this training-free method stands against explicitly trained compositional models.

---

> ### Author Rebuttal · Authors · 2026-03-31
>
> We sincerely thank you for reviewing our paper and for your constructive feedback. Below, we address your concerns in detail:
>
> > ### **W.1: LLM API request latency**
>
> Thanks for the question! The ∼5s latency from LLM-based layout planning is a small fraction of the end-to-end runtime and is unlikely to be a practical bottleneck. As shown in **Table A**, Wan2.2-A14B takes 411s on a single H100, so layout planning adds only 5/411≈1.2% overhead. In high-throughput scenarios, this cost can be further removed via pipelining: while generating the video for prompt A, we plan the layout for prompt B in parallel. Since generation is much slower than planning, the planning latency is almost hidden in practice.
>
> **Table A: Inference latency comparison on a single H100 GPU. LLM API latency is included.**
> | Model        | Wan2.2-A14B | + LVD | + ViCo | + Comp-Attn (Ours) |
> |--|--:|--:|--:|--:|
> | Latency (s) | 411         | 686   | 798    | 432    |
>
> > ### **W.2: Complex Physical Interactions**
>
> Thanks for raising this point. Comp-Attn is a training-free inference-time module, so its performance on complex physical interactions is largely bounded by the backbone model. It improves subject presence and inter-subject relation alignment, including in prompts with richer physical content, but it cannot guarantee strict physical consistency when the foundation model (e.g., Wan2.2-A14B) lacks strong understanding of fluid dynamics or collisions. We will clarify this limitation in the paper and expect it to diminish as future T2V foudation models improve their physical modeling ability.
>
> > ### **W.3 & Q.2: Sensitivity to hyperparameter and more ablation studies**
>
> We appreciate this question. The hyperparameter τ is selected empirically, and we find the method is not highly sensitive to the choices. We add an ablation on the temperature factor τ. As shown in **Table B**, varying the temperature leads to only minor performance differences, which supports the robustness of our method to this hyperparameter. This is expected since SCI primarily injects clearer semantics per subject, whereas τ only adjusts weight sharpness and does not change the vector injection direction. Averaging over diverse prompts further reduces apparent sensitivity.
>
> **Table B: More ablation results on the temperature factor.**
> | τ | 0.1 | 0.2 | 0.5 | 1.0 |
> |--:|--:|--:|--:|--:|
> | T2V-CompBench (Avg) |  0.6378|  0.6414|  0.6386|  0.6325|
>
> We further ablate the linear decay schedule in **Table C**. Removing it drops the overall T2V-CompBench score from 0.6414 to 0.6291, consistent with our claim in L200–L210 that stronger early-stage semantics injection helps ensure subject presence. The linear decay schedule is a useful optimization, but the main gain comes from SCI itself.
>
> **Table C: More ablation results on the linear decay schedule.**
> |  | w/o w(t)| Ours|
> |--:|--:|--:|
> | T2V-CompBench (Avg) |  0.6291|  0.6414|
>
> > ### **W.4: Comparison to training-based methods**
>
> We appreciate the suggestion. **Table D** compares Comp-Attn with the training-based, layout-controlled BlobGEN-Vid [1] on T2V-CompBench-v1 under the same CogVideoX-5B backbone. Comp-Attn is competitive and slightly better on average, supporting our view that layout control mainly improves inter-subject relations while subject omission can persist.
>
> **Table D: Comparison to training-based methods.**
> | Method | Con-Attr | Dyn-Att | Spatial | Motion | Num | Avg |
> |---|---:|---:|---:|---:|---:|---:|
> | BlobGEN-Vid [1] | 0.7400 | 0.2650 | 0.6725 | 0.3880 | 0.3910 | 0.4913 |
> | Comp-Attn |  0.7625 | 0.2716 |  0.6648 |  0.3869 |  0.4028 | 0.4977 |
>
> > ### **Q.1: Layout Sensitivity and Planner Latency**
>
> Thanks for the question. Layout-planning errors are rare in our setting (only ~1.1% prompts show noticeable motion/size errors; see Appendix). We further test sensitivity on T2V-CompBench by perturbing box layouts via random scaling (0.5×–1.5×) and randomly freezing box updates across adjacent frames. **Table E** shows only slight drops, indicating low sensitivity to moderate layout errors.
>
> **Table E: Layout sensitivity analysis.**
> | Method | Con-Attr | Spatial | Motion |
> |---|---:|---:|---:|
> | Wan2.2-A14B | 0.8394 | 0.6618 | 0.3654|
> | Comp-Attn | 0.8726 | 0.7244 | 0.4708|
> | Comp-Attn (w/ noise) |  0.8681 | 0.7162 |  0.4629 |
>
> Moreover, we can run layout planning with a smaller locally hosted model (e.g., Qwen3-4B). On a single H100 it takes ~3.5s, comparable to the API latency. We also observe no significant performance drop with the local planner (**Table F**). Thus both are viable. Local models improve deployability, while APIs are easier to scale and can be executed in parallel with generation (as discussed in W.1).
>
> **Table F: Different plannner analysis.**
> | Planner | T2V-CompBench (Avg) | Time |
> |---|---:|---:|
> |  Qwen3-Instruct-4B | 0.6359 | 3.56s|
> | GPT-4o-mini | 0.6414 | 5.22s|
>
> [1] BlobGEN-Vid: Compositional Text-to-Video Generation with Blob Video Representations, CVPR 2025

---

### Decision · Program_Chairs · 2026-04-30

**Decision:**

Accept (regular)

**Comment:**

Final Recommendations: All three reviewers (bzau, fPPr, and 6Cgj) gave the paper a consistent score of 4: Weak Accept. The paper is technically solid but retains a few minor weaknesses.

Primary Weaknesses & Post-Rebuttal Status:

LLM Layout Bottleneck & Hallucinations: Multiple reviewers questioned the reliance on GPT-4o-mini for spatial planning. The authors successfully argued that layout errors are rare (~1.1%), local models (like Qwen3-4B) can be swapped in, and the soft modulation easily handles moderate layout noise.

Presence vs. Relation Evaluation: Reviewer 6Cgj pointed out the lack of a direct metric separating "subject absence" from "misaligned relations." The authors resolved this during the rebuttal by running a new human and GPT-5 evaluation on 200 prompts, proving that SCI mostly helps presence while LAM mostly helps relations.